



# Ice nucleating particles measured in Swiss alpine snow samples are spatially, temporarily and chemically heterogeneous

Killian P. Brennan[1], Robert O. David[1,*], Nadine Borduas-Dedekind[1,2]

[1]Institute for Atmospheric and Climate Science, ETH Zurich, Zurich, 8092, Switzerland
[2]Institute for Biogeochemistry and Pollutant Dynamics, ETH Zurich, Zurich, 8092, Switzerland
*Now at Department of Geosciences, University of Oslo, Oslo, 0315, Norway

*Correspondence to*: Nadine Borduas-Dedekind (nadine.borduas@usys.ethz.ch), @nadineborduas

**Abstract.**
Ice nucleating particles (INPs) produce ice from supercooled water droplets through heterogeneous freezing in the atmosphere. Since the concentration of ice crystals affects the radiative properties of clouds as well as precipitation, constraining the liquid water to ice ratio could help reduce aerosol-cloud interaction uncertainties. INPs have been collected at the Jungfraujoch research station (at 3500 m a.s.l.) in central Switzerland; yet spatially diverse data on INP occurrence in the Swiss Alps are scarce and
remain uncharacterized. We address this scarcity through our Swiss Alpine snow sample study which took place during the winter of 2018. We collected a total of 88 fallen snow samples across the Alps at different locations, altitudes, terrains, times since last snowfall and depths. The INP concentrations were measured using the homebuilt DRoplet Ice Nuclei Counter Zurich (DRINCZ) and were then compared to spatial, meteorological and physiochemical parameters. We also extend an alternative way of
displaying frozen fraction (FF) versus temperature data through visualizing freezing temperatures as a boxplot to field collected samples. This plotting method displays the freezing temperature in one dimension, instead of the former two dimensions of FF vs temperature, allowing a condensed display of freezing temperature measurements. In the collected snow samples, large variability in INP occurrence was found, even for samples collected 10 m apart on a plain and 1 m apart in depth. Furthermore,
undiluted samples had INP concentrations ranging between 1 and 100 INP ml$^{-1}$ of snow water over a temperature range of −5 to −19 °C. From this field-collected data set, we parameterize the INP concentrations per milliliter of meltwater as a function of temperature with the following equation $c_{air}^*(T) = e^{-0.7T-7.05}$, comparing well with previously reported precipitation data presented in Petters and Wright, 2015.


When assuming a cloud water content of 0.4 g m$^{-3}$ and a critical INP concentration for glaciation of 10 m$^{-3}$, the majority of the snow precipitated from clouds with glaciation temperatures between −5 and −20 °C. Based on the observed variability in INP concentrations, we conclude that studies conducted at the high-altitude research station Jungfraujoch are representative for INP measurements in the Swiss
Alps. Furthermore, the INP concentration precipitation estimates allow us to extrapolate the concentrations to a cloud frozen fraction. Indeed, this approach for estimating the liquid water to ice ratio in mixed phase clouds compares well with aircraft measurements, ground-based lidar and satellite retrievals of cloud frozen fractions. In all, the generated parameterization for INP concentrations in meltwater could help estimate cloud glaciation temperatures.

**1. Introduction**

The frozen fraction of clouds significantly alters the radiative and dynamic properties of mixed-phase clouds. The abundance of ice crystals in a cloud can influence its optical transmittivity and reflectivity, as well as affect the formation of precipitation (Matus and L'Ecuyer, 2017; Mülmenstädt et al., 2015; Prenni et al., 2007; Vergara-Temprado et al., 2018). Since homogenous freezing of cloud droplets only
occurs at temperatures below −38 °C (Koop, 2004; Pruppacher and Klett, 2010; Stöckel et al., 2005), heterogeneous nucleation which involves the presence of ice nucleating particles (INPs) is required for





ice formation at warmer temperatures (Fletcher, 1962; Pruppacher and Klett, 2010; Vali et al., 2015). Aerosol-cloud interactions, including INP-cloud interactions, currently contribute to the largest uncertainty in total anthropogenic radiative forcing in global climate models (Boucher and Quaas, 2013) and hence constraining INP concentrations and better predicting the liquid water to ice ratio could help

reduce this uncertainty. Indeed, a growing number of INP parameterizations are helping to better constrain and predict INP concentrations as a function of temperature (DeMott et al., 2010; Petters and Wright, 2015; Phillips et al., 2008, 2012), surface area (Niemand et al., 2012) or organic carbon (Borduas-Dedekind et al., 2019; Wilson et al., 2015).

Mixed-phase clouds are responsible for the majority of land falling precipitation (Hande and Hoose, 2017; Henneberg et al., 2017; Mülmenstädt et al., 2015; Murray et al., 2012) through immersion freezing or by the freezing of cloud droplets due to the presence of an INP (de Boer et al., 2011). As the majority of ice precipitation in mixed-phase clouds is initiated via immersion freezing (de Boer et al, 2011), the composition, abundance and sources of INPs responsible for the precipitation can be

obtained through the analysis of precipitation samples (Fan et al., 2018; Morris et al., 2014; Stopelli et al., 2015; Wright et al., 2014). Indeed, INP concentrations have been measured in precipitation samples such as rain, snow, hail and sleet and have been recently revisited to obtain a representative INP concentration range for different temperatures (Petters and Wright, 2015). These samples have been collected in the United States (Hader et al., 2014; Hill et al., 2014; Petters and Wright, 2015; Wright et

al., 2014), Canada (Vali, 1966, 1971a), France (Joly et al., 2014), Switzerland (Stopelli et al., 2014, 2017) and more recently in Israel (Zipori et al., 2018). Furthermore, geographically diverse precipitation samples have shown consistent immersion freezing activity due to biological material (Christner et al., 2008a, 2008b). These field studies highlight the ubiquitous presence of INPs in precipitation while simultaneously emphasising high variability in INP concentrations.


Meanwhile, studies conducted in the laboratory or on non-precipitation samples conducted to further understand immersion freezing have found mineral dust, metallic oxides, and biological material to be efficient INPs (Hoose and Moehler, 2012; Kanji et al., 2017). Soluble INPs have also been shown to be efficient INPs if they contain extracts from plant-based material, including proteinaceous material

(Augustin et al., 2013; Dreischmeier et al., 2017; Koop and Zobrist, 2009; Pummer et al., 2012, 2015; Wilson et al., 2015). In addition, organic matter may also act as INP in the immersion freezing mode (Borduas-Dedekind et al., 2019; Hill et al., 2016; Hiranuma et al., 2015; Irish et al., 2019; Knopf et al., 2018; Wilson et al., 2015).

Field measurements of INPs in the Swiss Alps are largely bound to the high-altitude research station Jungfraujoch (Boose et al., 2016; Chou et al., 2011; Conen et al., 2012, 2017; Ehrman et al., 2001; Farrington et al., 2016; Hammer et al., 2018; Lacher et al., 2017, 2018a, 2018b; Lloyd et al., 2015; Meola et al., 2015; Mertes et al., 2007; Nillius et al., 2013; Stopelli et al., 2016, 2017) due to its typical free tropospheric conditions. As such, measurements of INPs in the free troposphere at Jungfraujoch

should be representative of INP concentrations globally. However, the impact of local sources and INPs present in the boundary layer cannot be ascertained. Therefore, to determine the representativeness of free tropospheric INP measurements across the Swiss Alps, geographically diverse sampling sites are required.

Thus, the first purpose of this study was to determine the heterogeneity of INP occurrence in the Swiss Alps and to investigate its correlation with various physicochemical properties and sampling site characteristics. 88 snow samples were collected using sterile Teflon tubes in the winter of 2018 across 17 locations on 15 different days spanning an approximate area of 35,752 km$^2$ in the Swiss Alps. INPs in the meltwater were measured in the immersion freezing mode using the Droplet Ice Nuclei Counter

Zurich (DRINCZ) (David et al., 2019). The ability of the snow meltwater to freeze is reported as freezing temperature in one-dimensional box plots. Furthermore, this study investigates INP occurrence by (1) location, (2) time, (3) altitude, (4) snow age, (5) total organic carbon, (6) conductivity and (7) INP size trough filtering. We further propose a constrained parameterization of INP as a function of temperature to complement the data set from (Petters and Wright, 2015).



The second purpose of this study was to extrapolate our measured INP concentrations to frozen cloud fractions within mixed-phase clouds over the Swiss Alps. We assumed a cloud water content of 0.4 g m$^{-3}$ (Petters and Wright, 2015) to determine the cumulative concentration of INPs in air and subsequently assumed a critical number of INPs for cloud glaciation, ranging between 10 and 1000 m$^{-3}$. From this information, glaciation temperatures of mixed-phase clouds and supercooled liquid water to ice ratio were estimated. Finally, we discuss the limitations of our approach for arriving at frozen cloud fractions as well as compare our results with aircraft measurements, ground-based lidar measurements, satellite retrievals and global circulation models (McCoy et al., 2016).

## 2. Methods

### 2.1. Sampling sites

Snow sampling locations were selected to provide a breadth of elevation, terrain type, distances from Jungfraujoch and snow age during one winter season to represent conditions in the Swiss alpine area during the hydrological winter of 2017/2018. The campaign took place between February and May 2018 and the sites were accessed by ski or by alpinism. All samples were taken from undisturbed top layer snow, unless it was a depth profile. Specifically, 17 sites throughout the Swiss Alps were chosen (Table 1 and Table S1). Two of these sites were within the boundaries of two different ski resorts, Davos (Weissfluh) and Andermatt (Sankt Annafirn), and the altitudes at which these two samples where obtained from are well above the altitude of artificial snow production. The coordinates of the sampling sites were measured using the GPS on a smartphone and span an area of 218 km from east to west and 164 km from north to south to cover an approximate area of 35 752 km$^2$. The altitudes of the sampling sites ranged from 440 to 3981 m above sea level (a.s.l.) with a median value of 2294 m a.s.l. (Figure 1), and therefore cover a wide range of altitudes within the Alps according to the (Bundesamt für Landestopografie swisstopo, 2018). Based on the date and time of sampling, snow age representing the timespan between the snowfall event and the sampling instance was determined by using the snowpack database (Institut für Schnee- und Lawinenforschung SLF, 2018), and more specifically by using the snowfall history at the SLF measurement station closest to the sampling site, typically within a few kilometers and at most within 10 km, and exceptionally within 25 km for Pilatus due to lack of stations in the Luzern region.

**Table 1: List of the dates, sites (altitude and longitude), altitude, location including the Swiss canton and number (n) of samples collected during this Swiss alpine snow sample study. Additional details with recorded parameters for each sample are available in the supplementary information.**

| Date | lat [°N] | lon [°E] | Altitude [m a.s.l.] | Location/Canton | n | notes |
|---|---|---|---|---|---|---|
| 04.02.18 | 46.68 | 7.40 | 1715 - 2033 | Schibe/Bern | 2 | |
| 13.02.18 | 46.53 | 7.48 | 2053 - 2316 | Rauflihore/Bern | 2 | |
| 24.02.18 | 47.04 | 9.11 | 2242 - 2294 | Schilt/Glarus | 4 | |
| 01.03.18 | 47.42 | 8.54 | 440 | Wahlenpark/Zurich | 2 | |
| 04.03.18 | 46.59 | 7.48 | 1945 | Meniggrat/Bern | 1 | |
| 17.03.18 | 46.55 | 7.46 | 1625 | Alpetli/Bern | 8 | Depth profile |
| 02.04.18 | 46.54 | 7.47 | 2045 | Chalberhöri/Bern | 9 | Depth profile |
| 14.04.18 | 46.44 | 7.57 | 1954 - 1967 | Engstligenalp/Bern | 6 | |
| 14.04.18 | 46.41 | 7.55 | 2185 - 3222 | Grossstrubel/Bern | 7 | |
| 14.04.18 | 46.44 | 7.56 | 1983 | Engstligenalp/Bern | 14 | Depth profile |
| 15.04.18 | 46.84 | 9.79 | 2822 | Weissfluh/ Graubünden | 9 | 10m homogeneity |
| 19.04.18 | 47.25 | 9.34 | 2416 | Säntis/Appenzell | 4 | |
| 22.04.18 | 46.17 | 7.99 | 3227 - 3981 | Fletschhorn/Valais | 3 | |
| 22.04.18 | 46.60 | 8.60 | 2729 | Sankt Annafirn/Uri | 6 | 10m homogeneity |
| 25.04.18 | 46.98 | 8.26 | 2020 | Pilatus/Obwalden | 2 | |





| 12.05.18 | 45.94 | 6.96 | 3137 - 3388 | Pte Aig. Verte/France | 4 | |
| 21.05.18 | 47.01 | 9.01 | 2772 | Vrenelisgärtli/Glarus | 4 | |
| **15 days of sampling** | **45.94 to 47.42 (164 km)** | **6.96 to 9.79 (218 km)** | **440-3981 (median at 2294 m)** | **17 locations** | **88 samples** | **3 depth profiles** |

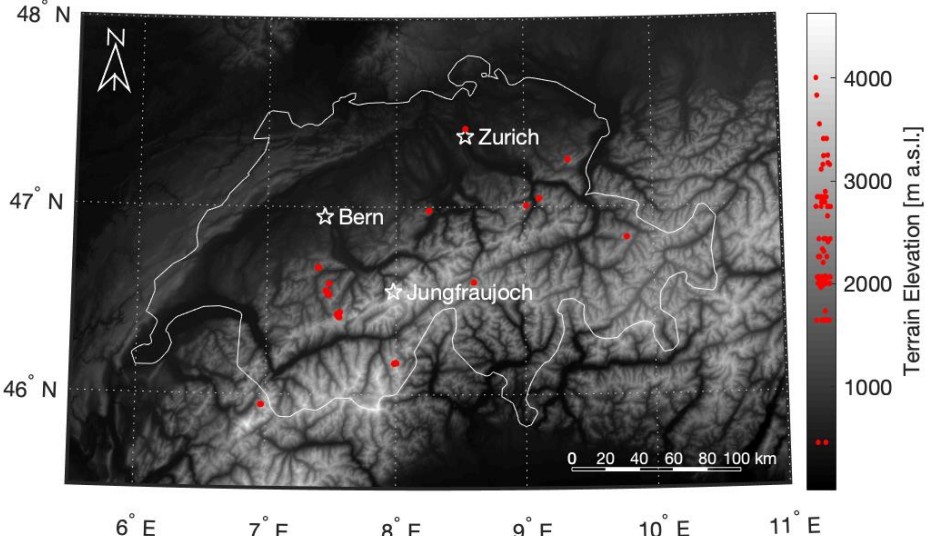

**Figure 1: Map of all sampling sites (red), of the Swiss border (white), and of the positions of the cities of Bern and Zurich, as well as the high-altitude alpine research station Jungfraujoch. The respective altitudes of the sampling sites are shown on the colour bar to the right, scattered in the x-direction to avoid overlapping of points. The terrain elevation is greyscale shaded according to a digital elevation model produced using Copernicus data and information funded by the European Union – EU-DEM layers (European Environment Agency, 2018).**

### 2.2. Sampling procedure

Previously reported snow collection procedures for INP studies have involved a Teflon-coated tray put in place before snowfall (Stopelli et al., 2017). This method is convenient and adequate for alpine research stations where appropriate accommodation is available. However, to assess the heterogeneity of INPs in the Alps, we needed to access remote locations. Collecting snow samples from the top of the snowpack allows sampling in multiple locations compared to other precipitation collection procedures, such as rainfall collection, which require sampling site infrastructure. Through ski-touring and alpine mountaineering, we accessed 17 different locations for one-time snow sampling. Indeed, no infrastructure was necessary and remote locations as high as 3981 m a.s.l. could be uniquely accessed.

Snow samples were collected in 50 mL polypropylene conical centrifuge tubes (Techno Plastic Products AG, Switzerland) without pre-treatment. In fact, preliminary pre-treatment testing showed degradation of the polypropylene material under heated conditions (125 °C in an oven) and when washed with organic solvents such as acetone, ethanol, methanol and acetonitrile as well as 0.1 M HCl (Figure S1). For every Techno Plastic Products tube batch, a tube was filled with 20 mL molecular biology reagent water (89079-460, Sigma Aldrich, USA) and placed in a freezer at −20 °C. These reference water samples were later measured alongside the snow samples to generate background water values for measurement comparison. The exact collection procedure for gathering snow samples is





described step by step in the supplementary information. Briefly, the collection was done by a skier or an alpinist wearing nitrile gloves and they ensured that only the tubes came in contact with the snow. The snow was compacted and the tubes were sealed. After collection, the snow samples were kept as cold as possible until the return to the laboratory, during which some of the snow did melt. Nonetheless,
refreezing experiments suggested little difference in frozen fraction (see Figure 3), allowing us to conclude minimal sample degradation during transport.

Snow depth profiles were dug with an aluminium avalanche rescue shovel and were dug until the underlaying dirt or rock surface was reached (see Figure 10 for a site photo). The depths of the samples
collected within the snowpack were determined with a graduated avalanche rescue probe (Mammut AG, Switzerland). Sample depths were distributed throughout the snowpack depending on the number of sampling tubes available on the specific trip.

### 2.3. Immersion freezing experiments and data analysis with DRINCZ

The DRoplet Ice Nuclei Counter Zurich (DRINCZ) technique used in this study, as well as the data processing and analysis are described in-depth by David et al. (2019). Additional details on the immersion freezing experimental details and on the frozen fraction (*FF*) analysis and can also be found in the supplemental information.

The instrument DRINCZ yields the *FF* from the number of wells frozen at a given temperature
($n_{frz}(T)$) and the total number of wells ($n_{tot} = 96$). A *FF* curve is a monotonic function of temperature and is derived according to Eq. (1). In other words, the frozen fraction stays the same or rises with decreasing temperatures.

$$FF(T) = \frac{n_{frz}(T)}{n_{tot}} \qquad (1)$$

The DRINCZ technique is limited to snow samples that freeze at temperatures above −22.5 °C, since
50% of the wells of the water background freeze at that temperature ($T_{50} = -22.5$ °C).

In order to extrapolate the *FF* determined by DRINCZ into an INP concentration, Poisson distribution calculations were used as in Eq. (2) (Vali, 1971b, 2018):

$$n_{mw}(T) = -\frac{1}{V_d}\ln(1 - FF(T)) \qquad (2)$$

where $n_{mw}(T)$ is the cumulative INP concentration per mL of meltwater as a function of temperature and $V_d$ is the droplet volume in mL (0.05ml) (Figure 2).

The detectable INP concentrations range of DRINCZ for the tested samples range from 0.6 to 150 mL[-1]. Note that we chose to report our correlation analysis using $T_{50}$ as the temperature where *FF* = 0.5, since
INP concentrations are not available for all temperatures. We discuss INP concentrations in the context of the overall dataset and for comparison with other published datasets. Although different arguments on omitting the first two wells exist (DeMott et al., 2016; Polen et al., 2018), we argue that trimming enhances representativeness, reproducibility and confidence in the INP concentrations. Thus, due to contamination potentially observed in the freezing of the first couple of wells at higher temperatures,
the first two wells to freeze were omitted for calculating INP concentrations. Furthermore, triplicates show good reproducibility, $T_{50}$ of sample triplicates fall within 1°C (Figure 3) and standard deviations did not depend on average freezing temperature, consistent with (Wright et al., 2013). Refreezing results show a similar spread in $T_{50}$ as the triplicates, which suggest that the IN activity of the samples is only minimally affected by freezing (Figure 3). Consistent with the observed spread in triplicate
freezing temperatures, the reported temperature uncertainty of DRINCZ is ± 0.9 °C (David et al., 2019). Finally, background corrections for the freezing temperatures were not necessary to report $T_{50}$ values, as all of these values were statistically significantly above the water background of the instrument. Only three samples have 75th percentile freezing temperatures overlapping with the mean of the background water: samples 21, 24 and 62. No further data manipulation was done for these samples





as the conclusions drawn from these freezing temperatures were the same with or without a correction (Table S1).

In addition to showing *FF* in a two-dimensional line graph (see Figure 2), we chose to show temperature dependent freezing events as a boxplot. This method reduces from two to one the required number of dimensions for plotting a single experiment. This visualization allows for the clear comparison of many samples side by side for every sample measured in this study and using all 96 data points without trimming (Figure S2). Polen et al. (2018) similarly uses boxplots to show results of several measurements for instrument development purposes and we are extending this approach to our 10   field collected data.

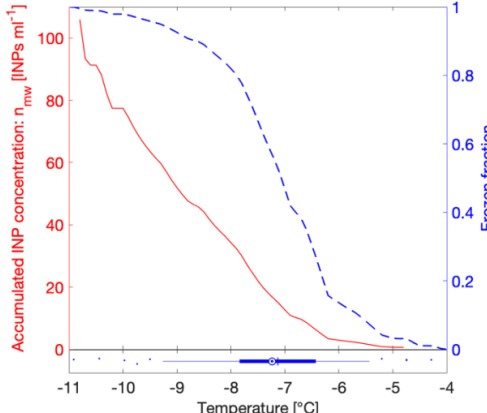

**Figure 2: Example of INP concentration generated from an *FF* curve. The left y-axis (red) shows the INP concentrations and the right y-axis (blue) represents the frozen fraction as a function of temperature. Below, the corresponding boxplot of the sample is** 15   **depicted. On the box plot, the thin blue vertical line shows the median and is equal to *T₅₀*, the mean is shown as a blue circle with a concentric dot. The blue box ranges from the 25th to the 75th percentile, whereas the whiskers extend from the 5th to the 95th percentile. Outliers are drawn as blue dots and scattered vertically to avoid overlapping.**

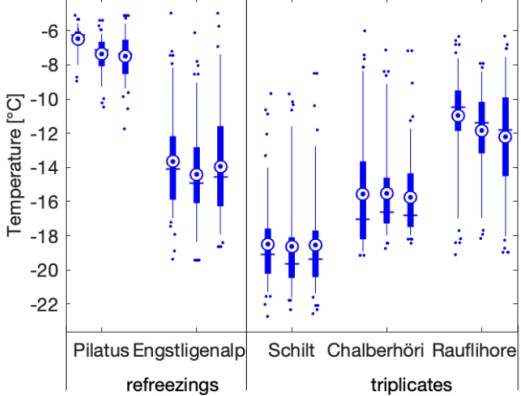

**Figure 3: Refreezing triplicate data shows that the variability in the refreezing (SD 0.47) is comparable to the variability in the** 20   **triplicate (SD 0.28) and well within the instrument error. On the box plots, the blue vertical line shows the median and is equal to T₅₀. The mean is shown as a blue circle with a concentric dot and the box ranges from the 25th to the 75th percentile. Finally, the whiskers extend from the 5th to the 95th percentile. Outliers are drawn as blue dots and scattered to avoid overlapping.**

### 2.4. Physicochemical analyses

The Swiss alpine snow samples were submitted to chemical analyses in an attempt to correlate 25   parameters to INP concentrations. In particular, total organic carbon (TOC), pH and conductivity





measurements were made for all samples, whereas filtering procedures were conducted for a subset of 17 samples.

The total organic carbon was quantified as the non-purgeable organic carbon (NPOC) in solution using a total organic carbon (TOC) analyser (TOC-L CSH, Shimadzu, Japan). This instrument uses a 680 °C combustion catalytic oxidation method, while providing quantification in water samples above 0.1 mg C L$^{-1}$. A detection limit of 4 µg C L$^{-1}$ is achieved through the use of a nondispersive infrared (NDIR) detector. Method sparge time and gas flow were 1.5 mins and 80 mL, respectively. For the NPOC calibration, two calibration solutions were prepared with concentrations of $20 \pm 0.2$ mg C L$^{-1}$ and $2 \pm 0.02$ mg C L$^{-1}$ from a potassium phthalate TOC standard solution of 1000 mg C L$^{-1}$ (Sigma-Aldrich).

pH values were measured with a Metrohm pH glass electrode and the values obtained were consistent across all samples, ranging between 5.2 and 6.2. Considering the error on the pH measurements from the unbuffered meltwater solutions, the pH of all the samples were within error and consequently was not ascribed to any INP variability.

The conductivity of the melted snow samples was measured using a handheld conductivity meter (LAQUAtwin COND, Horiba, Japan). It uses small sample volume (less than 0.5 mL) and has an accuracy of $\pm 1$ µS cm$^{-1}$ for conductivity measurements between 0 and 1999 µS cm$^{-1}$. The instrument was calibrated with a 1413 µS cm$^{-1}$ standard solution. Before the measurement was obtained, the sensor was washed three times with nanopure Milli-Q water and then 1 mL of the sample was flushed over the sensor for conditioning.

Finally, to classify the size of INPs in the snow samples, the samples were filtered through cellulose acetate membrane filters with pore sizes of 0.2 and 0.45 µm (514-0063, VWR, USA). The filtered samples were then measured with DRINCZ. Tests done with a 0.7 µm glass fiber filters (SF1300-07, BGB-Analytik, USA) reduced the freezing temperatures of the sample to lower temperatures than with the cellulose acetate filters. This result would lead to lower INP occurrence in measurements of 0.7 µm filtrates, versus filtrates of sizes 0.2 and 0.45 µm and hence the use of 0.7 µm glass fiber filters was discontinued for our study. In addition, samples were filtered with a 0.02 µm filter (Whatman® Anotop® syringe filters purchased through Sigma Aldrich) similarly to Irish et al, (2017). Lower $T_{50}$ values were observed with the 0.02 µm filtrates and we wondered whether salts from the filter could have been introduced into our sample, leading to a freezing point depression. However, the conductivity of SA water filtered through a 0.02 µm filter was measured at 4 µS cm$^{-1}$, a value too small to lead to a noticeable freezing point depression. We therefore concluded that the low freezing fractions observed with the 0.02 µm filtrates of the snowmelt was indeed due to loss of INPs in solution.

### 3. Results and discussion

This study investigated the freezing temperatures of 88 snow samples collected over 15 days and 17 different locations during the winter of 2018 in the Swiss Alps. The $T_{50}$ values and freezing temperatures are presented in three sections: physiochemical properties, times series and sampling site characteristics. $T_{50}$ values from the entire dataset ranged between −5.3 °C and −21.6 °C with a mean value of −12.5 ± 4.0 °C (Table S1), all above the DRINCZ instrument's detection limit. We first use $T_{50}$ values to compare samples to different physicochemical properties of the meltwater, such as TOC, conductivity and particle size, as well as to collection time within the winter season. We then use freezing temperature boxplots to compare the freezing behavior of samples based on terrain, altitude and snow age. Next, INP concentrations in precipitation were estimated and a parameterization is derived based on a temperature dependence. Finally, we extrapolate the INP concentrations to frozen cloud fractions to infer glaciation temperature and liquid water to ice ratio of mixed-phase clouds over the Swiss Alps in the winter of 2018.





**Table 2: Correlation analyses between T$_{50}$ values and the following physicochemical and terrain parameters. R$^2$ values characterize the linear relationship between T50 and the parameter; the p-value identifies whether the correlation is statistically significant; n represents the number of snowmelt samples used in the correlation.**

| Parameter | R$^2$ | p-value | n |
|---|---|---|---|
| Conductivity | 0.0015 | 0.722 | 88 |
| TOC | 0.0085 | 0.192 | 86 |
| Snow age | 0.0623 | 0.057 | 59 |
| Altitude | 0.0015 | 0.722 | 88 |
| Date | 0.0005 | 0.833 | 89 |
| Depth | 0.0028 | 0.624 | 88 |
| Distance | 0.2140 | 0.174 | 88 |

5       **3.1. Physicochemical properties**

   **3.1.1. Chemical properties**

To study the INP concentration dependence on chemical parameters, $T_{50}$ values were compared to total organic carbon (TOC) and conductivity measurements (Figure 4). Organic carbon values ranged between 0.3 and 5.2 mg C L$^{-1}$, with a median of 0.64 mg C L$^{-1}$ and an average of $0.9 \pm 0.7$ mg C L$^{-1}$.
The sample with the highest TOC value of 5.2 mg C L$^{-1}$ was collected at Grossstrubel in Valais at 2638 m a.s.l (Figure 4 and Table S1). Its $T_{50}$ value was −14.2 °C, and nothing was remarkably unique about this location. Furthermore, 13 snow samples were collected the same day with TOC values closer to the average concentration. Low TOC concentrations were found to span the entire range of $T_{50}$ values, indicating no correlation between TOC and freezing temperatures of the snowmelt. Wilson et al.
(2015) propose an INP parameterization for sea surface microlayer samples based on TOC, however our snow samples suggest no correlation with organic carbon content.

Conductivity is a measurement of ionic strength of the meltwater and salt can lower the freezing temperatures of a sample. However, conductivity values of the meltwater ranged from 0 to 14 µS cm$^{-1}$
with a median at 3 µS cm$^{-1}$ and an average of $4.1 \pm 2.5$ µS cm$^{-1}$. The three samples with the highest conductivity measurements were samples 88, 96 and 73 and were from Fletschhorn, Pointe Aiguille Verte and Weissfluh, respectively (Figure 4). These samples had TOC concentrations close to 1 mg C L$^{-1}$ and uncorrelated $T_{50}$ values. The conductivity values of this dataset are small, and close to the limit of detection of the conductivity meter, approximately representing salt concentrations below
400 µM of for example NaCl.

The pH values obtained were consistent across all meltwater samples, ranging between 5.2 and 6.2. This result is consistent with a pH of unbuffered pure water in equilibrium with atmospheric $CO_2$. The pH variability has no clear effect on IN ability and lies within the uncertainty of the measurement.
Note that the chemical properties of the meltwater are representative of all types of aerosols present within the collected snow sample. Yet, only a subset of these aerosols are INPs and can thus be related to the freezing temperatures. Since total aerosol concentrations are several orders of magnitude greater than INP concentrations (Kanji et al., 2017), the chemical signature of an INP is possibly lost among
total aerosol chemistry, even if the INPs are expected to be concentrated in precipitation samples. Therefore, it is not possible to develop a meaningful parameterization for INP concentrations based on TOC or conductivity from the meltwater samples collected in this study (Figure 4).



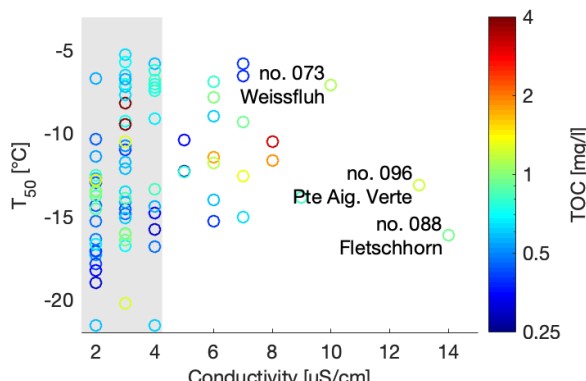

**Figure 4** Scatterplot of $T_{50}$ versus conductivity TOC is shown in colour and the three most conductive samples are labelled with the sample number and sampling site. The range of the conductivity instrument background is shaded in grey.

### 3.1.2. Meltwater filtration experiments

Determining the size of INPs may be useful in identifying their sources, transport and sedimentation pathways in the atmosphere. The meltwater was thus filtered to three sizes: 0.45, 0.2 and 0.02 µm (Figure 5). Filtrates from 0.45 µm and 0.2 µm filters retained the ice activity of the unfiltered sample. Three exceptions were observed from Figure 5; three of the highest $T_{50}$ values from the unfiltered meltwater lost between half to all their ice nucleating ability. However, filtrates from 0.02 µm filtrations

completely lost the ice activity of the original meltwater. Therefore, we conclude that most INPs in the collected snow samples were between 0.2 and 0.02 µm in size. This sizing test suggests that most of the INPs were found in the accumulation mode and smaller, and consequently only a few, albeit the most active, INPs were present in the coarse mode (0.5-5 µm) and larger. Sizes relevant for dry deposition are therefore only sparsely represented in our samples, suggesting that the majority of INPs found in the

meltwater were deposited in the snowpack through wet deposition by snow precipitation. This observation is expected and consistent with INPs preferentially removed by precipitation by initiating it (Stopelli et al., 2015). In addition, we make the same conclusion by plotting $T_{10}$ and $T_{50}$ which give the same trend in the size dependence of INPs (Figure S3). Although $T_{10}$ has been used in previous studies (Irish et al., 2017), $T_{50}$ was chosen here, as it was used in the rest of our analysis and as the same

conclusions can be drawn from both values.

The majority of INPs found within our samples were smaller than 0.2 µm in size, consistent with findings of existing studies on organic matter (Irish et al., 2017, 2019; Wilson et al., 2015). In contrast, Mason et al., 2016 found that the majority of INP particles had aerodynamic diameters in the coarse

mode when sampled from ambient air. However, it is important to note that it is not possible to determine the change in particle morphology when immersed in water and therefore the same INPs may be responsible for the observed ice nucleation activity in precipitation and air samples. Regardless, the size classification of INPs from this study provides further support for the limited role of bacteria in ice nucleation over the Swiss Alps, since bacteria are larger than 0.2 µm. However, the role of bacterial

fragments or proteinaceous material cannot be excluded (Hartmann et al., 2013; Pummer et al., 2012, 2015).





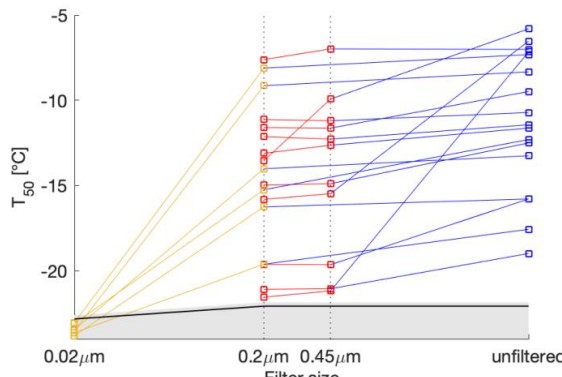

**Figure 5** $T_{50}$ **as a function of filter size for select samples. SA background** $T_{50}$ **is shown in black and the respective standard deviation is shaded in grey. The samples were filtered at 450, 200 and 20 nm and the lines connect the different filtrates of the same sample. The** $T_{10}$ **version of this graph can be found in the supplementary information for further comparison with (Irish et al., 2017; Wilson, 2015).**

### 3.2. Time series of $T_{50}$ values

The variability of $T_{50}$ values was observed throughout the measurement period and there was no trend over time (Figure 6). Noticeably, all samples taken in May 2018 had TOC values above the dataset average, and specifically above 1.2 mg C L$^{-1}$, however the time since their last snowfall was only 1-2 days and so these samples were still relatively fresh (Table S1). Nonetheless, the scatter of $T_{50}$ values indicate no trend overtime of the freezing temperatures of INPs, consistent with a lack of seasonality in INP concentrations measured by a continuous flow diffusion chamber at Jungfraujoch (Lacher et al., 2018a). The absence of time dependency on the INP occurrence within the campaign timeframe indicates that the variability is consistently large throughout the entire timespan investigated.

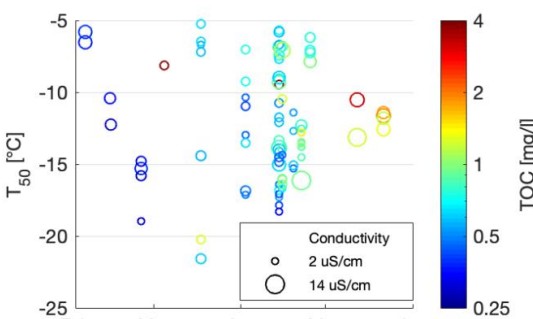

**Figure 6 Time series distribution of the** $T_{50}$ **values during the entire field campaign, displayed as months in 2018. TOC values of the samples are shown on a colour scale and conductivity values are shown on a marker size scale.**

### 3.3. Sampling site characteristics

#### 3.3.1. Spatial heterogeneity of freezing temperatures

To assess the local variability of INP concentrations spatially, samples from the same site, sampled on the same day were compared to their distance from each other (Figure 7). The sampling site at Schilt showed consistent freezing temperatures for samples separated by only 5 m, but showed a difference in $T_{50}$ of 4 °C for the sample located 300 m away, within a similar environment. Furthermore, the Sankt Annafirn site also display little variability of the median and mean freezing temperatures within a 5 m trajectory on top of the snow pack. Nonetheless, a variability in $T_{50}$ of 3 °C within 5 m was measured. Thus, we find that within a site of a radius of approximately 5 m, snow samples display similar INP concentrations. However, 300 m was enough to observe differences in INP concentrations.





To eliminate variability due to changes in altitude, we studied the spatial variability in INP concentrations at Engstligenalp due to its location in a flat flood plain of approximately $1 \times 2$ km in size (Figure S4). Due to the topographical homogeneity of the Engstligenalp site, similar local influences on
INP concentration were expected. However, within the plain of less than 2 km, a difference in $T_{50}$ of 8 °C was observed and even for samples collected a mere 10 m apart, a difference in $T_{50}$ of 2 °C was found (Figure 7). Additionally, samples taken on a snow hill side at the Weissfluh site, showed a similar wide spread in freezing temperatures within distances of 8 m. Both the Engstligenalp and Weissfluh sites displayed visual evidence of snowdrift, which could explain the large variability observed
compared to the Schilt and Sankt Annafirn sites (Figure 7). In fact, wind drift can lead to an extremely heterogeneous snowpack (Gauer, 2001), which could lead to locally heterogenous INP concentration. Specifically at Weissfluh, the origin sample and the samples at distances of 2 and somewhat at 4 m had significantly warmer $T_{50}$ temperatures and a narrower spread in freezing temperature than the other distances. This narrow spread suggests that the INPs responsible for the observed freezing in these
samples (origin and 2 m) were abundant, but inhomogeneous across the plains (Figure 7).

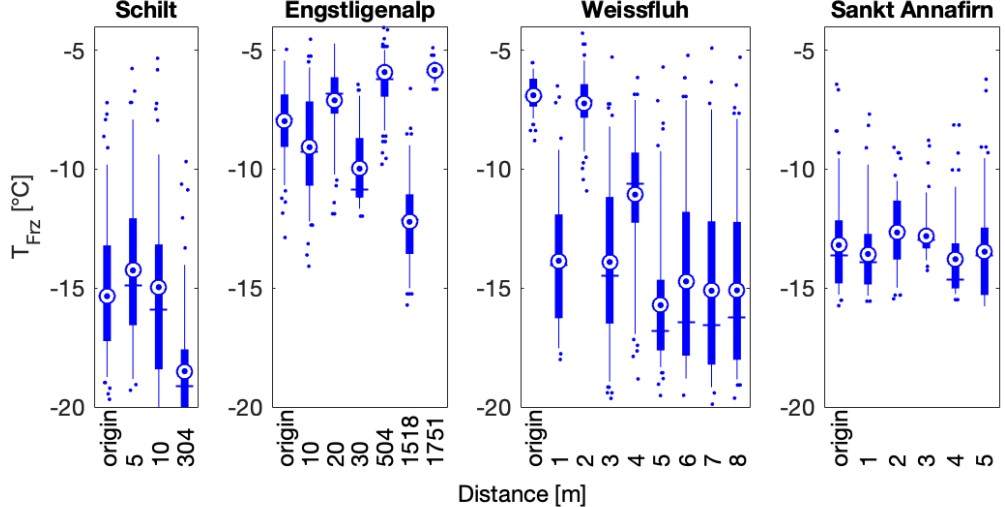

**Figure 7: Spatial distribution of freezing temperatures of snow samples collected at four different flat sites, at Schilt on February 24, 2018, at Engstligenalp on April 14, 2018, at Weissfluh on April 15, 2018, and at Sankt Annafirn on April 22, 2018. Weissfluh and Sankt Annafirn are particularly striking since the snow samples collected at these sites show heterogeneity within 1 m distances. All**
**the samples at the same location were collected on the same day. On the box plots, the blue vertical line shows the median and is equal to $T_{50}$, the mean is shown as a blue circle with a concentric dot and the box ranges from the $25^{th}$ to the $75^{th}$ percentile, and the whiskers extend from the $5^{th}$ to the $95^{th}$ percentile.**

### 3.3.2. Altitude dependency

To investigate the influence of boundary layer and local sources of INPs on snowfall across the Swiss
Alps, sampling sites were chosen to cover a broad range in altitude. The sampling sites ranged in altitude between 440 and 3981 m a.s.l. with a median altitude of 2294 m (Table 1), thereby covering a range of sites affected by rare snow events and eternal snowpack, respectively. When investigating the role of altitude on INP occurrence in the same sampling region there is a qualitative decrease in $T_{50}$ at higher altitudes for sites Fletschhorn and Pte Aig. Verte (Figure 8). The reasons are two-fold; first the
more active INP nucleate ice at warmer temperatures and are therefore removed earlier and lower in the clouds (Stopelli et al, 2015) or second, due to the presence of INP in the boundary layer that are advected into the cloud due to orographic lifting or turbulence at the top of the boundary layer. Indeed, (Lacher et al., 2018b) saw an increase in INP concentrations during periods of boundary layer air at Jungfraujoch, albeit at much colder sampling conditions than measured in this study. In contrast to an
altitude dependence observed at Fletschron and Pte. Aig. Verte, Grossrubel had no clear dependence on median and mean freezing temperatures with altitude, although samples at altitudes lower than 2500 m





had significantly narrower spreads of the 25th and 75th percentiles (Figure 8). When considering the observed spatial variability when sampling at distances over 5 m, it is difficult to disentangle the effect of altitude and of natural variability in INP occurrence. This dataset is therefore limited in its ability to compare free tropospheric snowfall and snowfall influenced by the boundary layer to assess the

importance of boundary layer aerosols on ice formation in clouds.

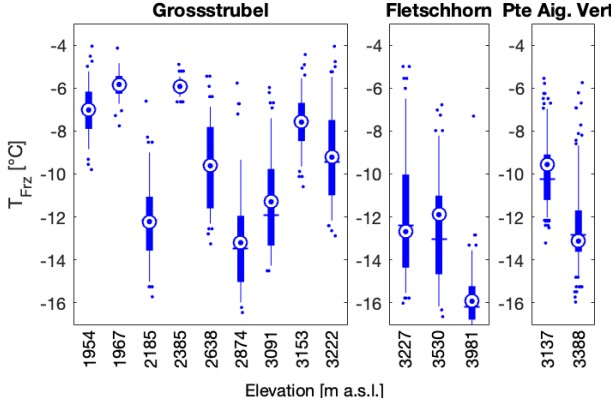

**Figure 8**: Snow samples were collected at different altitudes at Grossstrubel on April 14, 2018, at Fletschhorn on April 22, 2018, and at Pointe Aiguille Verte on May 12, 2018. On the box plots, the blue vertical line shows the median and is equal to $T_{50}$, the mean is shown as a blue circle with a concentric dot and the box ranges from the 25th to the 75th percentile, and the whiskers extend from
the 5th to the 95th percentile. Note that the two Pointe Aiguille Verte box plots contain 192 data points each as two individual snow samples were collected at the same location and thus combine for this figure.

### 3.3.3. Snow age through time and depth

The effect of snow age on the TOC, conductivity and freezing temperatures were evaluated using two types of snow sampling and data analysis. The first method involved classifying the $T_{50}$ temperatures of
the meltwater as a function of the day of the last snowfall reported at the nearest SLF station. The second method involved digging through the snowpack all the way to the ground and collecting samples along the transect (see photo in Figure 10).

Comparing different snowpack ages, which we define as the number of days between the most recent
snowfall at the nearest SLF site and the sampling day, allows the assessment of variables affecting snow properties since precipitation, including; (1) dry scavenging (Zhang et al., 2001), (2) microbial, plant and mineral effects from underlaying ground, (3) photochemical and ozone aging (Borduas-Dedekind et al., 2019; Gute and Abbatt, 2018; Kanji et al., 2013) (4), melt-process influence (Colbeck, 1981; Kuhn, 2001), (5) rainfall and (6) sublimation and deposition within the snowpack and between the snowpack
and the atmosphere (Birkeland et al., 1998). The correlation between snow age and $T_{50}$ was however found to be insignificant (Figure 9). However, the highest TOC value of 5.2 mg C L$^{-1}$ was observed for a meltwater sampled on April 4, 2018, nine days after the last snowfall which could have allowed the surface snow to concentrate (Figure 6, Figure 9). However, this hypothesis does not hold across the rest of the samples. Based on this result, we conclude that INP concentrations are not preferentially found in
fresh snow samples nor are they concentrated over time. This result is consistent with Hartmann et al. (2019) who found similar variability and concentrations of INPs in ice cores, which have much longer ageing times. However, it is important to note that we did not investigate the meteorological conditions between snowfall and sampling, nor did we measure the INP concentration at the same location on different days without any snowfall in between the days, although weathering has been shown to affect
the INP concentration in collected snow samples (Stopelli et al, 2014). Nonetheless, we can conclude that the role of ageing does not change the INP activity beyond the natural variability of the samples.

Sampling depth profiles in the snowpack permits a comparison of snow that precipitated at different times in the same location without the logistical burden of returning to the site several times during the
field campaign. Of note, the first widespread snowfalls that reached altitudes bellow 2000 m a.s.l. were





registered on November 6, 2017 and thus, the oldest snow and the deepest depth samples would have precipitated no earlier than this date. The depth samples in all three locations were very diverse, with $T_{50}$ values ranging from −20 to −5 °C (Figure 10). The depth profile sites of Alpetli and Chalberhöri were separated by only 1.5 km and were sampled 16 days apart, so one could have expected a similar

pattern in the depth profile if INP concentration were driven on the synoptic scale. However, no similarity nor trend were found as a function of depth at the three sites (Figure 10).

Furthermore, the Alpetli site had consistently warmer $T_{50}$ values at depths of more than 0.8 m from the surface of the snowpack in comparison to the other two sampling sites (Figure 10). Between 0 and 0.6

m beneath the snow at Alpetli however there was a wide spread in INP occurrence, and the freezing temperatures was similar to the boxplots of the other sites (Figure 10). The high activity of the INPs at Alpetli might be dominated by its particular proximity to trees shedding biogenic particles, known to efficiently nucleate ice (Morris et al., 2014). In contrast, the profiles at Engstligenalp and Chalberhöri had less variability in INP activity with depth and had a mean $T_{50}$ value of −13.1 and −12.7 °C,

respectively. Since the lower half of depth samples at the Alpetli site contained more efficient INPs in higher concentrations, we considered whether this high efficiency was due to contamination from the underlying soil or any organic matter within the soil, which has been shown to act as efficient INP in the immersion freezing mode (Hill et al., 2016; O'Sullivan et al., 2014; Suski et al., 2018; Tobo et al., 2014). As the depth of efficient INP extends well above the bottom of the profile (Figure 10) and as the

other two depth profiles do not display this behaviour, we conclude that contamination from the underlying soil is likely not responsible for the high $T_{50}$ values observed in the lower half of the Alpetli profile.

Although the age and exposure of the individual snow samples are unknown, there is no systematic

decrease in INP activity with depth, consistent with our snow age observations and with literature ice core measurements (Hartmann et al, 2019). Therefore, we argue that snow depth profile measurements are an appropriate method to analyze the concentration of INPs over a winter season. However, a study comparing the concentration of INPs in freshly fallen snow to that of the same corresponding snow in a depth profile is warranted. Ultimately, such a study would help to understand the representativeness of

the chronology within a snow depth profile, the influence of ageing on INPs and the impact of dry deposition on the concentration of INPs on a snow surface.

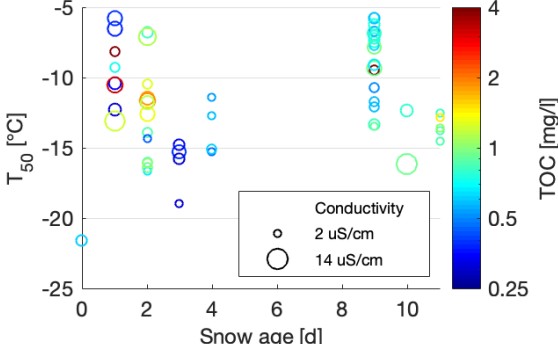

**Figure 9: Scatterplot of $T_{50}$ versus minimum snow age. TOC values of the samples are shown on a colour scale and the marker**
**surface area represent the conductivity.**





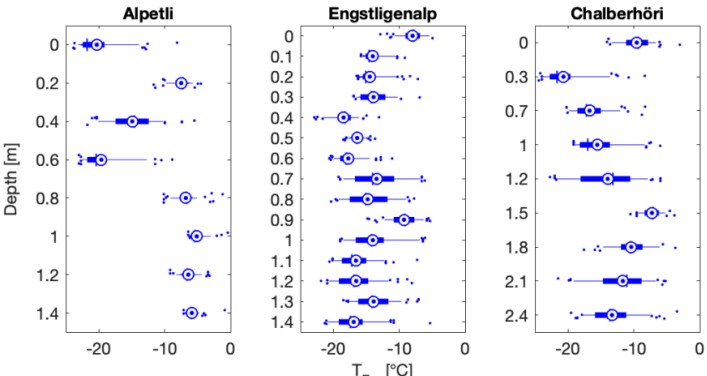

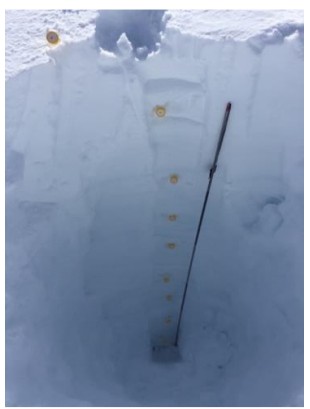

**Figure 10: Depth profiles of meltwater freezing temperatures at three different sampling sites, namely at Alpetli on March 17, 2018, at Engstligenalp on April 14, 2018 and at Chalberhöri on April 2, 2018 (Table 1). Right inset: a 2.7 m deep snow profile was dug with an aluminium avalanche rescue shovel at the sampling site Engstligenalp. The probe depicted is an avalanche rescue probe and**
**was used to measure the snow depth. The fresh layer snow was 9 days old and the first snowfall at that site was recorded on November 23, 2017. On the box plots, the blue concentric dot is the mean, the blue horizontal line is the median, the box is the 25th to the 75th percentile, and the whiskers extend from the 5th to the 95th percentile.**

### 3.4. INP concentration parameterization

Next, the freezing temperatures were converted into cumulative INP concentrations within the
meltwater ($n_{mw}$) following Eq. 2. The $n_{mw}$ values were extrapolated to cumulative INP concentrations in air ($c_{air}$) by assuming a cloud water content (CWC) of 0.4 g m$^{-3}$ (Petters and Wright, 2015) and the density of water of 1 g mL$^{-1}$ ($\rho_w$), according to Eq. 3

$$c_{air} = \frac{n_{mw} \times CWC}{\rho_w} \qquad (3)$$

A two-dimensional histogram of all INP concentrations was then obtained by binning the INP concentrations to a 1 °C bin-width laterally and a logarithmically constant bin-width vertically (Figure 11). The probability scale represents the probability of observing INPs within a given temperature and concentration bin based on all of our observations, including the depth profiles. The measured temperatures for a given INP concentration fall nicely within values found in previously published
studies (Petters and Wright, 2015), and furthermore within field-studies conducted at Jungfraujoch (Conen et al., 2012; Stopelli et al., 2014, 2016, 2017) (Figure 11).

The large variability in temperatures and INP concentrations made it difficult to find a clear influencing factor on INP occurrence, consistent with other field studies (Garimella et al., 2017; Lacher et al.,
2018a; Welti et al., 2018). Diverse sampling sites, in multiple regions, having varying environmental factors and their accompanying uncertainties complicates the determination of INP concentration predictive factors. Yet, Stopelli et al. (2016) found that upstream precipitation and wind speed accounted for 75% of the observed INP variability at the Jungfraujoch site. Unfortunately, these meteorological parameters could not be assessed in this study.
The large variability throughout the dataset (Figure 6) suggests that differing synoptic conditions influenced the snow samples at different timespans. Although contributing factors to INP occurrence are likely due to differing source regions and microphysical pathways upstream of the sampling locations, they could not be matched with physical or chemical properties (Table 2). Additionally, high
variability of snow sampled within close proximity of each other on the same day suggests that a site-specific contributing factor would be responsible. However, another explanation for local variability in INP occurrence when sampling fallen snow is the wind drift which might have transported snow during or after a precipitation event or might have uncovered an older layer, which was then later sampled next to a freshly precipitated layer. These complicating effects yield a heterogenous snowpack (Gauer,
2001), consistent with the measured heterogenous INP concentrations in this study. Nonetheless, the total measured INP concentrations fall well within the range of previously published values (Figure 11).





This study's dataset was used to generate a parameterization for potential applications in simulating INP concentrations in precipitation ($n_{nw}^*(T)$) and in air ($c_{air}^*(T)$) (Figure 11, Table 3). The parameterizations are given as $n_{nw}^*(T) = e^{-0.7T-6.02}$ for meltwater and $c_{air}^*(T) = e^{-0.7T-7.05}$ for air,

where $n_{nw}^*(T)$ and $c_{air}^*(T)$ are defined as the cumulative concentrations of INPs per mL of meltwater and per m³ of air, respectively, as a function of temperature (Table 3). In particular, the $c_{air}^*(T)$ parameterization can be directly applied to general circulation models in order to predict INP concentrations as a function of temperature.

Influence on cloud cover and precipitation through Saharan dust for example poses difficulties to numerical weather models in Europe (Knippertz and Todd, 2012), of which some portion is linked to uncertainty in INP activity. Due to the fact that INP occurrence in the measured snow samples have been determined as non-site specific, the studies conducted at Jungfraujoch, measuring the free troposphere are representative for INP assessments across the Swiss Alps.

**Table 3: Coefficients of the INP concentration parameterization, based on values between 0.2 and 90 INPs per mL and 0.08 and 36 m⁻³.**

| Function: $e^{a*T+b}$ | $a$ | $b_{mw}$ | $b_{air}$ |
|---|---|---|---|
| Mean | −0.70 | −6.02 | -7.05 |
| Lower bounds (mean-1 SD) | −0.60 | −6.81 | -7.84 |
| Upper bounds (mean+1 SD) | −0.96 | −5.84 | -6.87 |

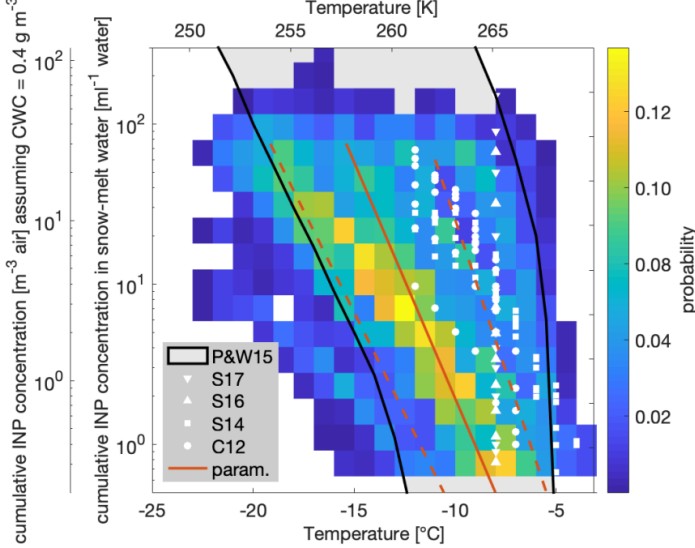

**Figure 11: Observations in INP concentrations in meltwater (n_mw) and in air (c_air) for all of the measurements as a function of temperature. The observations are shown as probability in a two-dimensional histogram. The grey area with black borders shows the concentrations found in the literature (Petters and Wright, 2015). The white symbols correspond to other field-studies in the Swiss alps (Conen et al., 2012; Stopelli et al., 2014, 2016, 2017). With the parameterization ($n_{mw}^*(T)/c_{air}^*(T)$) in red (solid) and 1σ standard deviations thereof (dashed).**

**4. Atmospheric implications**

**4.1. Mixed phase cloud activation**

An accurate temperature onset of ice nucleation in supercooled clouds and the subsequent transition to mixed-phase and eventually to glaciated clouds are important factors, yet sources of large uncertainties in weather and climate models through cloud lifetime and radiative forcing (Boucher and Quaas, 2013;

Prenni et al., 2007; Vergara-Temprado et al., 2018). As ice crystal concentrations in clouds can be




several orders of magnitude higher than INP concentrations (Wex et al., 2010), secondary ice processes can play an important in the evolution of supercooled clouds (Beck et al., 2018; Hallett and Mossop, 1974; Lauber et al., 2018; Mignani et al., 2019; Petters and Wright, 2015). Yet the process of ice multiplication is not fully understood, with multiplication factors ranging from one to multiple orders of magnitude (Mignani et al., 2019; Wang, 2013). Nevertheless, it has been proposed that complete glaciation of supercooled clouds can be initiated through ice multiplication with less than 10 m$^{-3}$ INPs ($c_{air}$) (Crawford et al., 2012; Mason, 1996) by the process of riming and ice splintering (Hallett and Mossop, 1974; Mossop, 1978). Using this assumption, we extrapolate our observations of $c_{air}$ to determine the temperature at which the clouds responsible for the collected snow samples would have been expected to glaciate by determining the temperature required for $c_{air}$ to exceed 10 m$^{-3}$ (Figure 12). As such, our snow samples indicate that the clouds would have been fully glaciated at temperatures above −25 °C and some even as high as −5 °C (Figure 12).

Furthermore, the glaciation temperature ($T_{glac.}$) of a cloud depends on the supersaturation within the cloud and is highly sensitive to updraft velocity (Korolev et al., 2017; Korolev and Isaac, 2003; Korolev, 2008). We extend our analysis by calculating the required temperatures for $c_{air}$ to exceed 10, 20, 40 and 400 m$^{-3}$ and we represent these results as frozen cloud fractions as a function of temperature (Figure 13). By rearranging Eq. 3, the INP concentration parameterization in Table 3, the temperature required for a certain concentration of INPs can be calculated as:

$$T_{glac.} = -(ln(\text{n}^*_{nw}) + 6.0)/0.7 \qquad (4)$$

By reorganizing Eq. 4, the change in temperature required for $c^*_{air}$ to increase by an order of magnitude can be calculated as $\Delta T = ln(10^{om})/0.7$ where $om$ is the order of magnitude in $\text{n}^*_{nw}(T)$. When $om = 1$, the change in temperature is equal to −3.3 °C. To extend our glaciation temperatures beyond 40 m$^{-3}$ and thus beyond the measured INP concentrations in this study, we can estimate $c_{air}$ at 400 m$^{-3}$ (Figure 13).

Our 50 % frozen cloud fractions fall within the range reported from airborne, ground based, and satellite measurements as summarized in McCoy et al (2016) (Figure 13). Furthermore, these results are remarkably consistent with the typically observed transition zones between supercooled liquid and ice clouds in models and observations (Costa et al., 2017; Henneberg et al., 2017; Lohmann et al., 2016; McCoy et al., 2016b; Pithan et al., 2014). Indeed, global circulation models partition liquid and ice in a given atmospheric volume as a monotonic function of temperature, but precipitation and freezing and melting cycles affect the liquid water to ice ratio, thereby modifying the cloud phase transition in often poorly constrained ways (Cesana et al., 2015; McCoy et al., 2015). Thus, temperature remains the dominant effect on influencing liquid cloud fraction (Tan et al., 2014).





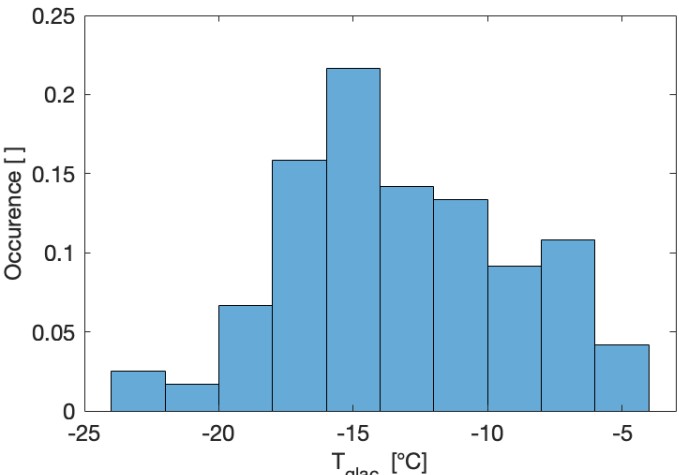

**Figure 12 Histogram of the cloud glaciation temperature (upper limit) assuming a $c_{air}$ value of 10 $m^{-3}$ (25 per mL of meltwater) probability occurrence as a fraction of one in the snow samples. Histogram bar-width is 2 °C.**

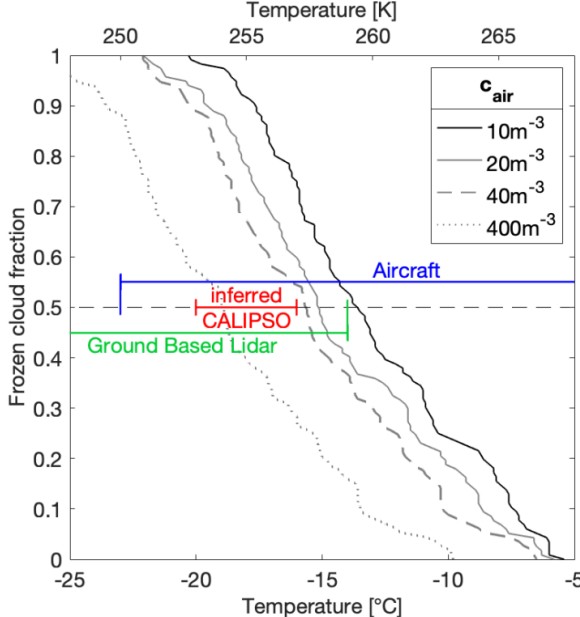

**Figure 13. Cloud frozen fraction curves as a function of temperature for different critical INP concentrations necessary for cloud glaciation. The reference temperature ranges for 50% frozen fractions are adapted from (McCoy et al., 2016a) and shown in color. The values for $c_{air}$ of 10, 20 and 40 $m^{-3}$ where obtained by means of concentration threshold from all collected data. In addition, the $c_{air}^*$ value for 400 $m^{-3}$ was estimated using Eq. 4 from $c_{air}^*$.**

## 4.2. Heterogeneity of ice nucleating particles in meltwater

The freezing temperatures of 88 individual snow meltwater samples taken at 17 different locations over 15 days of sampling are reported in this Swiss alpine snow sample study. There was no dependence of the freezing temperatures of the meltwater on the terrain or elevation. The snow age was evaluated through two methods: the first by the time since last snowfall measured at the nearest SLF station and





second by the snow depth profiles. Neither method led to any predictable trend in freezing temperatures, further highlighting the heterogeneity of INP concentrations across the dataset.

To relate previously reported precipitation measurements of INPs from the free troposphere at Jungfraujoch to the broader region of the Swiss Alps, the ice activities of the collected snow samples were analysed spatially. Within the limitations of the dataset, no site was found to exhibit particularly low or high freezing. Four sites where multiple snow samples were collected on the same day within a short distance of each other highlight that snowdrift can lead to high variability in freezing temperatures within a short distance. This result can be important when defining field sampling strategies for snow sample locations. Finally, based on the observed spatial variability in freezing temperatures and the fact that the meter scale variability is mainly caused by snowdrift, we conclude that studies conducted at the high-altitude research station Jungfraujoch are representative for INP assessments in the Swiss Alps.

**Code/Data availability**

Data available in the supplementary information. Raw data and MATLAB data analysis code available upon request.

**Competing interests**

The authors declare that they have no conflict of interests.

**Author contributions**

ROD and NBD designed the field study with contributions from KPB. KPB and NBD collected the samples. KPB conducted the experiments and analysed the data with contributions from ROD and NBD. The manuscript was written by all authors.

**Acknowledgements**

The authors acknowledge Robin Beglinger, Rafael Bonafini, Liam Brennan, Franz Friebel and Damian Urwyler for their expert ski touring and alpine mountaineering help in collecting snow samples. The authors thank Michael Rösch for technical assistance with DRINCZ and Ulrike Lohmann for insightful discussions. The work was in part financially supported by an SNSF Ambizione Grant (PZ00P2_179703) and by ETH Zurich.

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
