# Peer review of "Spatial and temporal variability in the ice nucleating ability of alpine snowmelt and extension to cloud frozen fraction"

_Atmospheric Chemistry and Physics, 2019_

## Short Comment (SC1) · 8 Aug 2019

The authors may want to consider referencing the recent paper by Creamean et al. (2019) whereby they report INP concentrations from aerosol, rime, and fresh fallen snow samples at Jungfraujoch:

Creamean, J. M., Mignani, C., Bukowiecki, N., and Conen, F.: Using freezing spectra characteristics to identify ice-nucleating particle populations during the winter in the Alps, Atmos. Chem. Phys., 19, 8123-8140, https://doi.org/10.5194/acp-19-8123-2019, 2019.

---

## Author Comment (AC1) · 8 Aug 2019

The title of the manuscript should read, "Ice nucleating particles measured in Swiss alpine snow samples are spatially, temporally and chemically heterogeneous".

To be clear, the word "temporarily" should have been "temporally".

Our sincerest apologies for this mistake.

Nadine Borduas-Dedekind

2019.

---

## Referee Comment (RC1) · Anonymous Referee #1 · 16 Aug 2019

In this manuscript Brennan et al. present results from INP measurements of snow samples collected at different locations, altitudes, times, and depth in the Swiss Alps. They found highly variable INP concentrations and used the data to generate a parameterization for the calculation of cloud glaciation temperatures.

The authors generated a very rich, unique and great data set of INP concentrations of 88 snow samples. This dataset is sound and the study is suited to the scope of the journal. The presented results are important for the ice nucleation community and can be useful for modelers. The experiments were well designed and were properly executed. I recommend publication after the following points have been addressed:

[Figure]

I do not understand how the authors come to the result and very prominent message presented in the title that the INP in the Swiss Alps are chemically heterogeneous. I cannot find experiments and results in the manuscript that support this message. Filtration of the samples just allows an estimation of the (physical) size of the INP. Determination of the pH, conductivity and TOC of the snow samples does not give information on the chemical composition of the INP within the snow.

In the abstract the authors write that they compared the INP concentrations with meteorological parameters. Which parameters were used? Where are the results?

Moreover, the authors highlight an alternative plotting method of INP data in the abstract. I wonder if there is another difference to Polen et al, 2018, than an extension to another sample type (field). Both studies used the same kind of data (freezing temperatures and frozen fractions). The sample type (field or laboratory) seems to be a secondary aspect, which does not explain why this is highlighted in the abstract.

Furthermore, the mixed use of "snow" (e.g., P7L25), "snowwater" (e.g., P1L25), "meltwater" (e.g., P8L26), "melted snow samples" (e.g., P7L18), "snow meltwater" (P2L50) and "snowmelt" (e.g., P8 Table 2 caption) for the same samples in this manuscript is confusing and leads to the impression that different types of samples have been analyzed e.g., P12L26/27 "..was observed for meltwater sampled on April 4,. . ." and 4.2 "Heterogeneity of ice nucleating particles in meltwater".

P2L28: "Soluble INPs have also been shown to be efficient INPs if they contain extracts from plant-based material,. . ." Please correct this statement. A soluble INP cannot contain "extracts". Moreover, for example fungal INP can nucleate at higher temperatures than plant INP (see cited reference Pummer et al 2015, ACP).

P4L18: Please add here the information that the tubes were sterile.

P5L3: The authors should either add some information on how the snow was compacted and/or refer to the supplement where this information is given. Did the authors

check with so called "handling blanks" that no contaminations occurred during sampling, compaction, transport and further handling of the tubes?

P5L8: How did the authors avoid cross-contamination when taking the depth profile samples? Was the shovel cleaned between the different sampling sites? How was made sure that no surface or "upper layer" INP were brought into the lower snow layers during shoveling?

P5L35: Why are INP concentrations not available for all temperatures? A short explanation should be added here.

P5L38: Can the authors explain why the first two frozen wells were considered as contaminations? I would expect a higher risk of contamination with lower T INP if there is a contamination as shown by the three samples which overlap with the water control.

P6L8: The authors write that they used the "data without trimming" although it was described in the paragraph before (P5L40) that the data were trimmed and the first two wells were excluded. This is confusing and needs clarification.

P6L8: Please cite the final paper of Polen et al. (published in AMT in Sep 2018). Polen, M., Brubaker, T., Somers, J., and Sullivan, R. C.: Cleaning up our water: reducing interferences from nonhomogeneous freezing of "pure" water in droplet freezing assays of ice-nucleating particles, Atmos. Meas. Tech., 11, 5315-5334, https://doi.org/10.5194/amt-11-5315-2018, 2018.

Figure 3: This figure and caption is not clear. Two locations are displayed as refreezing's and three other locations as triplicates. If refreezing and triplicates are different things one should compare the same samples/locations. Based on the caption "refreezing triplicate data" refreezings and triplicates seem to be actually the same i.e., refreezing of triplicates? Please clarify. Omit "Finally" in the caption.

P7L27: Please clarify. In L26 it is said that filter with pore sizes 0.2 and 0.45 $\mu$m were used to determine the size of the INP. But then results of a 0.7 $\mu$m filtration and

0.02 are presented too. It would help to restructure the paragraph about the filtration experiments and put the filtrations in a more logical order. Please correct "Tests done with a 0.7 $\mu$m glass fiber filters". Omit the "a" or the "s" from "filters".

P7L32: Omit "purchased through" - superflous

P7L35: Please introduce "SA water" when used first.

Table 2: " and the following...". Nothing follows, thus omit.

P9L21: Omit "in size". "smaller than 0.2 $\mu$m" is sufficient.

P9L29: It would help to mention the sources of such proteinaceous INP such as fungi and plant pollen (see cited reference Pummer et al. 2015, ACP).

Figure 5: "filter size"$\rightarrow$ "pore size", missing spaces before $\mu$m, " were filtered at 450 , 200 and 20 nm"..." $\rightarrow$ stay with $\mu$m, "filtered through";

P10L19: The header "Sampling site characteristics" sounds more like a description of the sampling sites and thus does not fit to the subsections and results presented in this section.

Figure 7/8: Is there a difference between TFrz as used here and Temperature used in Figure 3, which is the same type of plot?

Figure 13: I am not a cloud expert, but wonder if it would be better to use "frozen fraction of could droplets" instead of "frozen cloud fraction" as the INP concentrations are known.

P17L10: The subsection "4.2. Heterogeneity of ice nucleating particles in meltwater" is more or less redundant to the results and discussion already presented in "3.3. Sampling site characteristics" and seems not fit as a subsection in "4. Atmospheric implications". The sections 3.3. and 4.2 should be merged in 3.3 with a new heading, as suggested above.

Figure S1: Based on the information given in the method section (P4L23) water from Sigma Aldrich was used as control. What is, with only one type of water, the difference between the untreated tube and SA (I assume this means Sigma Aldrich water) reference? Moreover, the results of the methanol, acetonitrile and HCl washings as listed on P4L21 are not presented in Figure S1, but instead there is a Milli-Q wash presented. Figure and corresponding text should be checked for consistency and completeness.

Figure S2: "The grey shaded area..." In my version there is no grey shaded area in this figure. The authors might want to check this.

Figure S3: "Size dependency" -> "size determination"; "filter size" -> e.g., "pore size", " were filtered at 450 , 200 and 20 nm"..." → stay with $\mu$m, "filtered through"; omit " The T10 version..." as this is the T10 version.

Technical corrections

There are numerous other typos and inconsistencies throughout the manuscript. Many of those errors should have been caught by careful proofreading. I list the issues that caught my eye but I advise the authors to recheck their manuscript carefully to catch all of those typos.

Mixed use of l and L for Liter in text and figures. Mostly L is used but l is used in P1L15, Fig 2, P5L31, Fig 4, Fig 6, Fig 9.

Missing spaces (Table 1 – 10m, P5L31-0.05ml, Fig 5, Fig 13, Table S1, SuppP5L7, Fig S3).

Inconsistent format/symbols: T50 (Text vs Table S1, caption Table 2), $\mu$ (Text, Fig 5, Fig S3) vs uS (Fig 6, Fig 9).

Figures have partly axes with °C and K (Fig. 11, 13), partly only with °C (Fig 12).

Axis title capitalization (Fig 13) vs not capitalized (Fig 11).

Table 1: The "altitude" within the brackets should be "latitude".

P8L31: "for" all types.

Figure 4 caption: "conductivity TOC"→ "conductivity. TOC"?

P10L25: "display"->"displays".

Figure 12: Emty [] on the y-axis.

---

## Referee Comment (RC2) · Anonymous Referee #2 · 10 Sep 2019

In their study Brennan et al. investigate INP concentrations from snow samples taken in the Swiss Alps. In total 88 samples were collected during the winter of 2018. Samples were obtained from 17 locations covering a vast area of the Swiss Alps. Attention was paid to terrain characteristics, elevation, snow age, snow depth and distance to Jungfraujoch. INP concentrations were determined in the lab together with other physicochemical parameters of the bulk meltwater. Based on the INP concentration per ml meltwater the authors also create a parameterisation to calculate cloud glaciation temperature.

The study is well conducted and scientifically sound. Sampling procedures are de-

scribed in detail in the supplement material and are appropriate. To measure INP concentrations the authors use a newly developed method that has been tested and is extensively discussed in a recent publication referred to in the manuscript. While the analysis of the data is rather descriptive, a large benefit of the study is that samples were taken at 17 different locations in Switzerland spanning a large area. Most other studies focus on single locations. Sampling was also done from snow depth profiles and the local variability was investigated. Such data are very useful because they are rare.

Overall the manuscript provides a large dataset that is very beneficial for the ice nucleation community. The study fits the scope of ACP and I recommend publication after the points below have been addressed.

General comments

Title – The title should be changed. The current title suggests that the nature of INPs was studied. However, I don't see that much about the nature of INPs can be said from the study. E.g. chemical analyses were done on bulk samples and as the authors point out I agree that "the chemical signature of an INP is probably lost among total aerosol chemistry" (P834-35). The INP size was addressed but actually seems to be within the same range for all samples, so not heterogeneous. Consider addressing the variability of INP concentrations in the title rather than the INPs directly.

Why did the authors not calculate differential freezing nucleus spectra? They present a very useful picture of the entire INP population (Vali 2019) and INPs could be qualitatively classified (warm mode and cold mode INP, see also Creamean et al. 2019). Looking only at $T_{50}$ values might disguise the presence of a few INP at high temperatures. Comparing $T_{50}$ values is a rather limited approach when investigating heterogeneous environmental samples. The authors should consider adding freezing curves to the supplement material. Their current data representation in form of box plots looks nice but omits potentially relevant information.

[Figure]

Specific Comments

P1L19-23: The abstract seems rather long. I suggest that the authors cut out L19-23. This describes just another way of plotting data (box plots). I don't see why this is really novel. See also my general comment.

P1L19: As far as I can see meteorological parameters were not used. Delete "meteorological".

P1L28: The equation stated refers to c_air and not the INP concentration per ml meltwater. Please correct.

P3L19: The elevation of both sites is about 2800m. Why is this well above the altitude of artificial snow production?

P4 Figure1: Please color code the sampling locations by altitude. At the moment it is not possible to attribute a certain altitude to a specific location, which would be useful and can be easily added.

P4L32: First, I am not familiar with biology reagent water. What is the purpose of biology reagent water? Is there a difference to ultrapure (MilliQ) water? Please explain. Second, what was done with the so determined background values? Were they subtracted from the respective snow sample freezing curves?

P6L4-10: This is a nice idea but I think this approach is not ideal for field samples with most likely heterogeneous INP populations. See also my general comment.

P6L7-8: Here it is stated that the data was not trimmed and all 96 data points are used, while the previous paragraph explains that the data was trimmed (omitting 2 wells). This is confusing.

P6L22: add "horizontally" . . .and scattered "horizontally" to avoid overlapping.

P7L35-37: I wonder whether blank measurements were done with all filters? Figure 5 suggests so. Add this information here.

P10L1-5: "SA background T_50" What is SA?

P11L14-15: I don't understand the conclusion that INP were abundant but inhomogeneously spread. Is there evidence for less snow drift at the St. Anna Firn site? Did the authors compare wind speeds during the last snow fall at the sites?

P11 Paragraph "Altitude Dependence": In order to evaluate the influence of the boundary layer airmass trajectories should be analyzed. A site at e.g. 2000m can be in or out of the boundary layer depending on meteorological conditions.

P14L32-34: I don't see in what way source regions or microphysical pathways upstream of the sampling locations were analyzed, but the statement suggests so. Neither meteorological data nor airmass trajectories were included.

P17L10: This section reads more like "Conclusions" and should not be a subsection to "4. Atmospheric implications".

References

Vali, G.: Revisiting the differential freezing nucleus spectra derived from drop-freezing experiments: methods of calculation, applications, and confidence limits, Atmos. Meas. Tech., 12, 1219–1231, https://doi.org/10.5194/amt-12-1219-2019, 2019

Creamean, J. M., Mignani, C., Bukowiecki, N., and Conen, F.: Using freezing spectra characteristics to identify ice-nucleating particle populations during the winter in the Alps, Atmos. Chem. Phys., 19, 8123–8140, https://doi.org/10.5194/acp-19-8123-2019, 2019

---

## Author Comment (AC2) · 5 Nov 2019

Dear Jessie Creamean,

We thank you very much for sharing your recent work with us. It is indeed relevant to our study, and we sincerely apologize for the omisison of this reference to our original manscuript. We have now included your reference in our examples of reports of INP concentrations in precipitation samples at JFJ as well as in further comparing our plotting methods: boxplots vs differential INP for warm mode identification.

Don't hesitate to get in touch with us for further discussions.

Sincerely and all the best,

Nadine Borduas-Dedekind
* * *

---

## Author Response (AR1)

**Editorial Note:**
Referee comments are in black.
Author replies are in blue.
Text modified are in red.

**Anonymous Referee #1**

In this manuscript Brennan et al. present results from INP measurements of snow samples collected at different locations, altitudes, times, and depth in the Swiss Alps. They found highly variable INP concentrations and used the data to generate a parameterization for the calculation of cloud glaciation temperatures.

The authors generated a very rich, unique and great data set of INP concentrations of 88 snow samples. This dataset is sound and the study is suited to the scope of the journal. The presented results are important for the ice nucleation community and can be useful for modelers. The experiments were well designed and were properly executed. I recommend publication after the following points have been addressed:

We thank the reviewer for their support and critical feedback.

I do not understand how the authors come to the result and very prominent message presented in the title that the INP in the Swiss Alps are chemically heterogeneous. I cannot find experiments and results in the manuscript that support this message. Filtration of the samples just allows an estimation of the (physical) size of the INP. Determination of the pH, conductivity and TOC of the snow samples does not give information on the chemical composition of the INP within the snow.

We thank the reviewer for their comment. We agree that the chemical aspect of our study is rather homogeneous instead of heterogenous and that the title was therefore ambiguous. We show that the pH, conductivity and TOC of snow samples which are chemical indicators for hydronium ions, for inorganic ions and for organic carbon, respectively are more or less constant across our samples. We agree with the reviewer that size filtration is a physical property.

We have modified the title to ,"Spatial and temporal variability in the ice nucleating ability of alpine snowmelt and extension to cloud frozen fraction".

In the abstract the authors write that they compared the INP concentrations with meteorological parameters. Which parameters were used? Where are the results?

Good point. We made a mistake. The word "meteorological" was replaced by "temporal".

Moreover, the authors highlight an alternative plotting method of INP data in the abstract. I wonder if there is another difference to Polen et al, 2018, than an extension to another sample type (field). Both studies used the same kind of data (freezing temperatures and frozen fractions). The sample type (field or laboratory) seems to be a secondary aspect, which does not explain why this is highlighted in the abstract.

We would like to advocate for the use of one-dimension visualization for freezing temperatures to avoid typically over crowded frozen fraction plots. Yet, we agree that this mention does not belong in the abstract, and have therefore removed this section. In addition, we have reworded the sentences to remove any implied novelty. The sentence in the abstract now reads,

"Boxplots of the freezing temperatures show large variability in INP occurrence, even for samples collected 10 m apart on a plain and 1 m apart in depth."

Furthermore, the mixed use of "snow" (e.g., P7L25), "snowwater" (e.g., P1L25), "melt-water" (e.g., P8L26), "melted snow samples" (e.g., P7L18), "snow meltwater" (P2L50) and "snowmelt" (e.g., P8 Table 2 caption) for the same samples in this manuscript is confusing and leads to the impression that different types of samples have been analyzed e.g., P12L26/27 "..was observed for meltwater sampled on April 4,. . ." and 4.2 "Heterogeneity of ice nucleating particles in meltwater".

The reviewer makes a very valid point and we thank them for bring it up. To help clarify, we have opted to systematically use "snowmelt" throughout the manuscript.

P2L28: "Soluble INPs have also been shown to be efficient INPs if they contain extracts from plant-based material,. . ." Please correct this statement. A soluble INP cannot contain "extracts". Moreover, for example fungal INP can nucleate at higher temperatures than plant INP (see cited reference Pummer et al 2015, ACP).

The word "soluble" can certainly be ambiguous in the field of atmospheric ice nucleation. For the specific case of P2L28, we have decided to omit the word soluble and have reworded the sentence as,

"Efficient INPs may also originate from extracts of plant-based material, including proteinaceous material and polysaccharides (Augustin et al., 2013; Dreischmeier et al., 2017; Koop and Zobrist, 2009; Pummer et al., 2012, 2015; Wilson et al., 2015)."

P4L18: Please add here the information that the tubes were sterile.

Done.

P5L3: The authors should either add some information on how the snow was compacted and/or refer to the supplement where this information is given. Did the authors check with so called "handling blanks" that no contaminations occurred during sampling, compaction, transport and further handling of the tubes?

We thank the reviewer for their clarification request. In the supplementary information, we had written that "The snow was scooped directly with the open tube, to avoid contaminating the sample with hands or spatulas. If the snow was fresh, loose powder, it was necessary to compress the snow within the tube and sample multiple times to obtain a large enough sampling volume (optimal 30 mL). This compression was accomplished by repeatedly scooping snow and banging the bottom of the tube against a firm surface (for example ski boot)." In the main text, we added that the snow was "inertially" compacted.

To address the point of a handling blank, we added the following sentence,

"Since 3 of the collected 88 snowmelt samples had freezing temperature distributions at the water background of DRINCZ (Figure S2), we can suggest that no to little contamination could have been introduced during the snow sampling."

P5L8: How did the authors avoid cross-contamination when taking the depth profile samples? Was the shovel cleaned between the different sampling sites? How was made sure that no surface or "upper layer" INP were brought into the lower snow layers during shoveling?

For further clarification, we have added the following sentence,

"The side of the freshly dug hole was then scraped with the shovel to expose untouched snow at all depths and to remove possible cross-contamination from the ground during the digging process."

P5L35: Why are INP concentrations not available for all temperatures? A short explanation should be added here.

We agree with the reviewer that this comment was very confusing. We meant to say that for example T90 temperatures were not available for all samples, since they were recorded at the instrument background. We chose to delete that clause for clarity, since the word "detectable" was already present in the previous sentence. In the end, this section was reworded and rearranged, and in the end, this sentence no longer appeared in the text.

P5L38: Can the authors explain why the first two frozen wells were considered as contaminations? I would expect a higher risk of contamination with lower T INP if there is a contamination as shown by the three samples which overlap with the water control.

The reviewer is correct to question our discussion of "contamination", and we agree that contamination is more likely a problem at lower temperatures in absolute terms. We use all 96 raw data points for our presentation of freezing temperatures and boxplot, but for the data processing, in other words, from the raw data to cumulative INP concentrations, we chose to trim the first two wells to enhance reproducibility. We therefore make a distinction between raw data with no trimming, and INP concentrations with trimming. The discussion of section 2.3 was rewarded and reorganized to add clarity. It now reads,

"In addition to showing *FF* versus temperature in a two-dimensional line graph (blue line in **Error! Reference source not found.**), we show all 96 raw data points as freezing temperatures in a boxplot (bottom half of **Error! Reference source not found.**). The blue box ranges from the $25_{th}$ to the $75_{th}$ percentile of freezing temperatures, whereas the whiskers extend from the $5_{th}$ to the $95_{th}$ percentile. Within the boxplot, the median, equal to *T$_{50}$*, is shown as a thin perpendicular blue line to the box and the mean is shown as a blue circle with a concentric dot (**Error! Reference source not found.**). When the mean and the median values overlap, the FF curve is more or less linear (Figure 2 - left). However, when the mean and median values differ by one degree or more, the FF curve has a kink or bump in its slope (Figure 2 - right), often observed for biological INPs (Creamean et al., 2019). The boxplot graphing method reduces from two to one the required number of dimensions for displaying a single experiment. This visualization allows for the clear comparison of many samples side by side for every sample measured in this study and uses all 96 data points without trimming (Figure S2).

In order to extrapolate the *FF* determined by DRINCZ into an INP concentration, Poisson distribution calculations were used as in Eq. (2) (Vali, 1971b, 2019):

$$n_{sm}(T) = -\frac{1}{V_d}\ln(1 - FF(T)) \hspace{4cm} (2)$$

where $n_{mw}(T)$ is the cumulative INP concentration per mL of snowmelt as a function of temperature and $V_d$ is the droplet volume in mL (0.05 mL) (**Error! Reference source not found.**). Although different arguments on omitting the first two wells exist (DeMott et al., 2016; Polen et al., 2018), we argue that trimming enhances representativeness, reproducibility and confidence in the processing of the data from FF to INP concentrations. Thus, the first two wells to freeze out of the 96 wells were omitted for calculating cumulative INP concentrations."

P6L8: The authors write that they used the "data without trimming" although it was described in the paragraph before (P5L40) that the data were trimmed and the first two wells were excluded. This is confusing and needs clarification.

We agree with the reviewer that those two decisions are apparently confusing. We think the confusion could have come from our unclear order of the two paragraphs (the one discussed in this point and at point P5L38 above.) For clarity, we have moved the discussing of the raw data before the discussion of the processed data. We thank the reviewer for helping us better explain these steps, and refer the reviewer to the comment above for the final version of this section.

P6L8: Please cite the final paper of Polen et al. (published in AMT in Sep 2018). Polen, M., Brubaker, T., Somers, J., and Sullivan, R. C.: Cleaning up our water: reducing interferences from nonhomogeneous freezing of "pure" water in droplet freezing assays of ice-nucleating particles, Atmos. Meas. Tech., 11, 5315-5334, https://doi.org/10.5194/amt-11-5315-2018, 2018.

It is now correctly cited.

Figure 3: This figure and caption is not clear. Two locations are displayed as refreezing's and three other locations as triplicates. If refreezing and triplicates are different things one should compare the same samples/locations. Based on the caption "refreezing triplicate data" refreezings and triplicates seem to be actually the same i.e., refreezing of triplicates? Please clarify. Omit "Finally" in the caption.

We have clarified the titles in Figure 3 as well as the caption, as there was indeed an error and we thank the reviewer for noticing this error. The caption should have read refreezing "and" triplicate data at different locations. To further clarify, we have also divided the figure into part A) and part B). The caption now reads,

"Refreezing and triplicate data at different locations show that the variability is within the instrument error of 0.9 °C. The standard deviation of the refreezings (± 0.47 °C) is comparable to the standard deviation of the triplicates (± 0.28 °C). On the box plots, the blue vertical line shows the median and is equal to $T_{50}$. The mean is shown as a blue circle with a concentric dot and the box ranges from the 25th to the 75th percentile. The whiskers extend from the 5th to the 95th percentile."

P7L27: Please clarify. In L26 it is said that filter with pore sizes 0.2 and 0.45 μm were used to determine the size of the INP. But then results of a 0.7 μm filtration and 0.02 are presented too. It would help to restructure the paragraph about the filtration experiments and put the filtrations in a more logical order. Please correct "Tests done with a 0.7 μm glass fiber filters". Omit the "a" or the "s" from "filters".

We agree with the reviewer that this paragraph is confusing. We have rewritten it for clarity as follows,

"Finally, to classify the size of INPs in the snow samples, the samples were filtered through cellulose acetate membrane filters with pore sizes of 0.2 and 0.45 μm (514-0063, VWR, USA). In addition, samples were filtered with a 0.02 μm pore filter (Whatman® Anotop® syringe filters, Sigma Aldrich) similarly to Irish et al, (2017). The filtered samples, including molecular biology reagent water blanks, were then measured with DRINCZ. Tests with glass fiber filters with pose sizes of 0.7 μm (SF1300-07, BGB-Analytik, USA) yielded lower freezing temperatures than with the cellulose acetate membrane filters and were thus not used further for this study."

P7L32: Omit "purchased through" - superflous

Done.

P7L35: Please introduce "SA water" when used first.

We have removed the use of SA water, and call it molecular biology reagent water instead throughout the manuscript and supplementary information.

Table 2: " and the following. . .". Nothing follows, thus omit.

Done.

P9L21: Omit "in size". "smaller than 0.2 μm" is sufficient.

Done.

P9L29: It would help to mention the sources of such proteinaceous INP such as fungi and plant pollen (see cited reference Pummer et al. 2015, ACP).

Agreed. We have modified the sentence to read,

"However, the role of bacterial fragments or proteinaceous material from sources such as fungi and plant pollen cannot be excluded (Hartmann et al., 2013; Pummer et al., 2012, 2015)."

Figure 5: "filter size"→ "pore size", missing spaces before μm, " were filtered at 450 , 200 and 20 nm". . ." → stay with μm, "filtered through";

All changed.

P10L19: The header "Sampling site characteristics" sounds more like a description of the sampling sites and thus does not fit to the subsections and results presented in this section.

Agreed. We have renamed this heading, "Spatial, altitudinal, snow age and depth variability of freezing temperatures".

Figure 7/8: Is there a difference between TFrz as used here and Temperature used in Figure 3, which is the same type of plot?

No there is no difference. To clarify, we have changed the y-axis in Figure 3 to $T_{Frz}$.

Figure 13: I am not a cloud expert, but wonder if it would be better to use "frozen fraction of could droplets" instead of "frozen cloud fraction" as the INP concentrations are known.

We thank the reviewer for their comment. Unfortunately, without the droplet number within the cloud, we cannot say anything about the number (and thus the fraction) of cloud droplets which have frozen. The frozen cloud fractions were calculated from our 88 snowmelt samples. To further clarify Figure 13 (now updated to Figure 12 in the revised text), the section has been rewritten (although this revision was not directly prompted by the reviewer, we felt it was quite unclear as it was written).

P17L10: The subsection "4.2. Heterogeneity of ice nucleating particles in meltwater" is more or less redundant to the results and discussion already presented in "3.3. Sampling site characteristics" and seems not fit as a subsection in "4. Atmospheric implications". The sections 3.3. and 4.2 should be merged in 3.3 with a new heading, as suggested above.

Agreed. We completely deleted section 4.2. And names section 3.3 to the Spatial, altitudinal, snow age and depth variability of freezing temperatures.

Figure S1: Based on the information given in the method section (P4L23) water from Sigma Aldrich was used as control. What is, with only one type of water, the difference between the untreated tube and SA (I assume this means Sigma Aldrich water) reference? Moreover, the results of the methanol, acetonitrile and HCl washings as listed on P4L21 are not presented in Figure S1, but instead there is a Milli-Q wash presented. Figure and corresponding text should be checked for consistency and completeness.

We thank the reviewer for their comment and we apologize for our mistake and ambiguity in the name of the samples. We have also relabeled Figure S1 and avoided any mention of Sigma Aldrich water and instead label is as molecular biology reagent water. We have modified the text on P4L21 and in Figure S1's caption.

The text now read;

"In fact, preliminary pre-treatment testing showed physical degradation of the polypropylene material under heated conditions (125 °C in an oven) and when washed with organic solvents such as acetone and ethanol (Figure S1). For every Techno Plastic Products tube batch, a tube was filled with 20 mL molecular biology reagent water (89079-460, Sigma Aldrich, USA) and placed in a freezer at −20 °C (untreated tube samples in Figure S1). These reference water samples were later measured alongside the snow samples to generate background water values for measurement comparison."

The caption now reads, "Figure S1: Comparison of freezing temperature boxplots of molecular biology reagent water (89079-460, Sigma Aldrich, USA) stored (1 day plus freeze/thaw cycle) in Techno Plastic Products tubes after different washing procedures as well as and untreated tubes. The background water refers to water directly from the molecular biology reagent water bottle without any contact with the untreated tube."

Figure S2: "The grey shaded area. . ." In my version there is no grey shaded area in this figure. The authors might want to check this.

We thank the reviewer for the heads up. It seems there was an error during the conversion from Word to PDF. We have rectified the problem.

Figure S3: "Size dependency" -> "size determination"; "filter size" -> e.g., "pore size", " were filtered at 450 , 200 and 20 nm". . ." → stay with µm, "filtered through"; omit " The T10 version. . ." as this is the T10 version.

Fixed.

Technical corrections

There are numerous other typos and inconsistencies throughout the manuscript. Many of those errors should have been caught by careful proofreading. I list the issues that caught my eye but I advise the authors to recheck their manuscript carefully to catch all of those typos.

We thank the reviewer for their careful evaluation of our manuscript. We really appreciate it.

Mixed use of l and L for Liter in text and figures. Mostly L is used but l is used in P1L15, Fig 2, P5L31, Fig 4, Fig 6, Fig 9.

We have changed all instances to the symbol of "L".

Missing spaces (Table 1 – 10m, P5L31-0.05ml, Fig 5, Fig 13, Table S1, SuppP5L7, Fig S3).

All fixed.

Inconsistent format/symbols: T50 (Text vs Table S1, caption Table 2), µ (Text, Fig 5, Fig S3) vs uS (Fig 6, Fig 9).

Fixed.

Figures have partly axes with °C and K (Fig. 11, 13), partly only with °C (Fig 12).

Figures 11, and 12 now all have both °C (bottom) and K (top) temperature axes.

Axis title capitalization (Fig 13) vs not capitalized (Fig 11).

Both y-axes have now been capitalized in figure 11.

Table 1: The "altitude" within the brackets should be "latitude".

Fixed.

P8L31: "for" all types.
On this point unfortunately, we disagree with the reviewer. "Note that the chemical properties of the snowmelt are representative of all types of aerosols present within the collected snow sample" and not "for all types".

Figure 4 caption: "conductivity TOC"→ "conductivity. TOC"? P10L25: "display"->"displays".

Fixed.

Figure 12: Emty [] on the y-axis.

We meant to leave the brackets empty to show there were no units to this value. Since it wasn't clear, we have removed the brackets.

**Anonymous Referee #2**

In their study Brennan et al. investigate INP concentrations from snow samples taken in the Swiss Alps. In total 88 samples were collected during the winter of 2018. Samples were obtained from 17 locations covering a vast area of the Swiss Alps. Attention was paid to terrain characteristics, elevation, snow age, snow depth and distance to Jungfraujoch. INP concentrations were determined in the lab together with other physicochemical parameters of the bulk meltwater. Based on the INP concentration per ml meltwater the authors also create a parameterisation to calculate cloud glaciation temperature.

The study is well conducted and scientifically sound. Sampling procedures are described in detail in the supplement material and are appropriate. To measure INP concentrations the authors use a newly developed method that has been tested and is extensively discussed in a recent publication referred to in the manuscript. While the analysis of the data is rather descriptive, a large benefit of the study is that samples were taken at 17 different locations in Switzerland spanning a large area. Most other studies focus on single locations. Sampling was also done from snow depth profiles and the local variability was investigated. Such data are very useful because they are rare.

Overall the manuscript provides a large dataset that is very beneficial for the ice nucleation community. The study fits the scope of ACP and I recommend publication after the points below have been addressed.

We sincerely thank the reviewer for their positive feedback and critical assessment of our work.

General comments

Title – The title should be changed. The current title suggests that the nature of INPs was studied. However, I don't see that much about the nature of INPs can be said from the study. E.g. chemical analyses were done on bulk samples and as the authors point out I agree that "the chemical signature of an INP is probably lost among total aerosol chemistry" (P834-35). The INP size was addressed but actually seems to be within the same range for all samples, so not heterogeneous. Consider addressing the variability of INP concentrations in the title rather than the INPs directly.

We thank the reviewer for their insightful comment, and we agree that the title misrepresented our study. We have modified the title as follows, "Spatial and temporal variability in the ice nucleating ability of alpine snowmelt and extension to cloud frozen fraction".

Why did the authors not calculate differential freezing nucleus spectra? They present a very useful picture of the entire INP population (Vali 2019) and INPs could be qualitatively classified (warm mode and cold mode INP, see also Creamean et al. 2019). Looking only at T_50 values might disguise the presence of a few INP at high temperatures. Comparing T_50 values is a rather limited approach when investigating heterogeneous environmental samples. The authors should consider adding freezing curves to the supplement material. Their current data representation in form of box plots looks nice but omits potentially relevant information.

We thank the reviewer for their comment. We believe we didn't explain graphically and within the text the value of the boxplot method adequately. To further clarify the value of freezing temperatures as boxplots as alternatives to differential freezing nucleus spectra, we have remade Figure 2 and written a new paragraph in the methods section to clarify the interpretation of the freezing temperature boxplots. We believe that in the revised manuscript we now explicitly state the comparison with Creamean et al. 2019 as well as better highlight the value of the boxplots to depict warm modes and cold modes.

Revised Figure 2:

[Figure]

Revised text in the methods section:

"In addition to showing *FF* versus temperature in a two-dimensional line graph (blue line in **Error! Reference source not found.**), we show all 96 raw data points as freezing temperatures in a boxplot (bottom half of **Error! Reference source not found.**). The blue box ranges from the 25th to the 75th percentile of freezing temperatures, whereas the whiskers extend from the 5th to the 95th percentile. Within the boxplot, the median, equal to $T_{50}$, is shown as a thin perpendicular blue line to the box and the mean is shown as a blue circle with a concentric dot (**Error! Reference source not found.**). When the mean and the median values overlap, the FF curve is more or less linear (Figure 2 - left). However, when the mean and median values differ by one degree or more, the FF curve has a kink or bump in its slope (Figure 2 - right), often observed for biological INPs (Creamean et al., 2019). The boxplot graphing method reduces from two to one the required number of dimensions for displaying a single experiment. This visualization allows for the clear comparison of many samples side by side for every sample measured in this study and uses all 96 data points without trimming (Figure S2).

In order to extrapolate the *FF* determined by DRINCZ into an INP concentration, Poisson distribution calculations were used as in Eq. (2) (Vali, 1971b, 2019):

$$n_{sm}(T) = -\frac{1}{V_d}\ln(1 - FF(T))$$
(2)

where $n_{mw}(T)$ is the cumulative INP concentration per mL of snowmelt as a function of temperature and $V_d$ is the droplet volume in mL (0.05 mL) (**Error! Reference source not found.**). Although different arguments on omitting the first two wells exist (DeMott et al., 2016; Polen et al., 2018), we argue that trimming enhances representativeness, reproducibility and confidence in the processing of the data from FF to INP concentrations. Thus, the first two wells to freeze out of the 96 wells were omitted for calculating cumulative INP concentrations."

Specific Comments

P1L19-23: The abstract seems rather long. I suggest that the authors cut out L19-23. This describes just another way of plotting data (box plots). I don't see why this is really novel. See also my general comment.

We agree with the reviewer and we have edited this section in addition to shortening the overall abstract length from 412 words to 351 words.

P1L19: As far as I can see meteorological parameters were not used. Delete "meteorological".

Agreed. Done.

P1L28: The equation stated refers to c_air and not the INP concentration per ml melt- water. Please correct.

The reviewer is correct. We modified the expression to, "cumulative concentrations of INPs per m$_{-3}$ of air".

P3L19: The elevation of both sites is about 2800m. Why is this well above the altitude of artificial snow production?

These samples were collected where no evidence (as far as we can tell) of snow making was observed during the ski day at those resorts. We clarified our statement which now reads,

"Two of these sites were within the boundaries of two different ski resorts, Davos (Weissfluh) and Andermatt (Sankt Annafirn). Considering the altitudes at which these samples where obtained as well as an absence of operating snow canons during the sampling days, we expect to not have sampled any artificial snow."

P4 Figure1: Please color code the sampling locations by altitude. At the moment it is not possible to attribute a certain altitude to a specific location, which would be useful and can be easily added.

We thank the reviewer for their recommendations and have modified Figure 1 to add more information using an altitude colour scale. We have also added site labels to the Figure. We think the figure is much improved now in relating information about the sampling sites.

Revised Figure 1:

[Figure]

P4L32: First, I am not familiar with biology reagent water. What is the purpose of biology reagent water? Is there a difference to ultrapure (MilliQ) water? Please explain. Second, what was done with the so determined background values? Were they subtracted from the respective snow sample freezing curves?

Molecular biology reagent water is high purity water purchased from Sigma-Aldrich. In our method validation and technique optimization in DRINCZ (See David et al., AMTD, 2019), we obtained better reproducibility with the water from Sigma-Aldrich then with MilliQ water. We wonder if the Milli-Q water quality is affected by the lifetime of the UV lamp inside the instrument or by the warm up time of the lamp when the dispenser is turned off. In any case, the molecular biology reagent water continues to be a more reliable background for our drop freezing measurements.

No further corrections were made with the background. We did not subtract them from our obtained freezing temperatures, as that mathematical operation would not be consistent with Poisson statistics. We added a sentence to the text to clarify;

"Finally, background corrections for the freezing temperatures were not necessary, as all of the $T_{50}$ values were statistically above the water background of the instrument. Only three of the 88 samples had 75th percentile freezing temperatures overlapping with the mean of the background water: samples 21, 24 and 62 (Table S1). No further data manipulation was done for these samples as the conclusions drawn from these freezing temperatures were the same with or without a correction (Table S1)."

P6L4-10: This is a nice idea but I think this approach is not ideal for field samples with most likely heterogeneous INP populations. See also my general comment.

We appreciate the reviewer's comment. We think we did not do an effective job at communicating the value of a boxplot to display freezing temperatures. In an effort to better

communicate the boxplot display of freezing temperatures, we have rewritten this paragraph almost entirely as well as moved the paragraph sooner in the section. We have also modified Figure 2 to give the boxplot more prominence. By doing so, we hope to have better communicated its value, as we argue that no to little information is lost in the boxplot when one considers the difference between the freezing temperatures of the median and of the mean. In other words, "warm mode" can still be identified in the boxplots as it has been identified with differential freezing spectra in Creamean et al., ACP, 2019.

"In addition to showing *FF* versus temperature in a two-dimensional line graph (blue line in **Error! Reference source not found.**), we show all 96 raw data points as freezing temperatures in a boxplot (bottom half of **Error! Reference source not found.**). The blue box ranges from the $25_{th}$ to the $75_{th}$ percentile of freezing temperatures, whereas the whiskers extend from the $5_{th}$ to the $95_{th}$ percentile. Within the boxplot, the median, equal to $T_{50}$, is shown as a thin perpendicular blue line to the box and the mean is shown as a blue circle with a concentric dot (**Error! Reference source not found.**). When the mean and the median values overlap, the FF curve is more or less linear (Figure 2 - left). However, when the mean and median values differ by one degree or more, the FF curve has a kink or bump in its slope (Figure 2 - right), often observed for biological INPs in a so-called warm mode (Creamean et al., 2019). The boxplot graphing method reduces from two to one the required number of dimensions for displaying a single experiment. This visualization allows for the clear comparison of many samples side by side for every sample measured in this study and uses all 96 data points without trimming (Figure S2)."

P6L7-8: Here it is stated that the data was not trimmed and all 96 data points are used, while the previous paragraph explains that the data was trimmed (omitting 2 wells). This is confusing.

We agree with the reviewer that this section was misleading. To clarify, we used all 96 raw data points for our presentation of freezing temperatures and boxplot, but for the data processing, in other words, from the raw data to cumulative INP concentrations, we chose to trim the first two wells to enhance reproducibility. We therefore make a distinction between raw data with no trimming, and INP concentrations with trimming.

The text was modified accordingly,

"Although different arguments on omitting the first two wells exist (DeMott et al., 2016; Polen et al., 2018a), we argue that trimming enhances representativeness, reproducibility and confidence in the INP concentrations. Thus, the first two wells to freeze out of 96 were omitted for calculating INP concentrations."

P6L22: add "horizontally" . . .and scattered "horizontally" to avoid overlapping.

Done.

P7L35-37: I wonder whether blank measurements were done with all filters? Figure 5 suggests so. Add this information here.

Yes, molecular biology reagent water was used with all the filters and we have modified the sentence as follows,

"The filtered samples, including molecular biology reagent water blanks, were then measured with DRINCZ"

P10L1-5: "SA background T_50" What is SA?

SA means Sigma-Aldrich water background. The use of this acronym is ambiguous, and so we have now changed all instances of background water mentions, to specifically state that it is molecular biology reagent water instead.

P11L14-15: I don't understand the conclusion that INP were abundant but inhomogeneously spread. Is there evidence for less snow drift at the St. Anna Firn site? Did the authors compare wind speeds during the last snow fall at the sites?

Unfortunately, we do not have wind speeds at any of the sites. We are not looking for sources, but rather we wanted to study the variability, and look for correlations with different parameters. Wind speed might not be a clear indicator of blowing snow. For clarity we reworded the concluding sentence of this paragraph as,

"The narrow spread suggests that the INPs responsible for the observed freezing in these samples (at origin and at 2 m) were at abundant at these locations, but inhomogeneous across the plain (**Error! Reference source not found.**)."

P11 Paragraph "Altitude Dependence": In order to evaluate the influence of the boundary layer, airmass trajectories should be analyzed. A site at e.g. 2000m can be in or out of the boundary layer depending on meteorological conditions.

Airmass trajectories required to analyze the height of the boundary layer would require high resolution not available with HYSPLIT (0.25 degrees equivalent to roughly 25 km resolution has only be available since June 2019 and only 0.5 degrees is available for our sampling days in the winter of 2018). We are concerned about the impact of the terrain not being well represented in such a coarse model. To better answer the reviewer, we did HYSPLIT trajectories and the results are below. Unfortunately, we are unable to draw conclusions from these analyses as to whether the airmasses were in or out of the boundary layer. To further complicate this type of analysis, the time between the snowfall (recorded for these back trajectories in Figure RC1) and the sampling sites varied.

[Figure]

Figure RC1: HYSPLIT backward trajectories were generated at the last snowfall at the site using GDAS at 0.5 degrees resolution. The model parameters employed included model vertical velocity for the year, month, day and hour of sampling at Gorssstrubel (14.04.2019), Fletschhorn (22.04.2019), and Pointe Aiguille Verte (12.05.2019). Total run time was 84 hours at heights of 1000 m, 2000 m and 3000 m AMSL.

No further change to the text was made.

P14L32-34: I don't see in what way source regions or microphysical pathways up-stream of the sampling locations were analyzed, but the statement suggests so. Neither meteorological data nor airmass trajectories were included.

We acknowledge that we didn't look into HYSPLIT data. Because of resolution constraint, and because we saw high variability in the INP data.

P17L10: This section reads more like "Conclusions" and should not be a subsection to "4. Atmospheric implications".

Agreed. We have deleted section 4.2 since it was redundant and since ACP doesn't have a conclusion section.

50  representativeness, reproducibility and confidence in the processing of the data from FF to INP

$n_{mw}(T) = -\frac{1}{V_d}\ln(1 - FF(T)) \cdots\cdots (2)$¶
where $n_{mw}(T)$ is the cumulative INP concentration per mL of snow meltwater as a function of temperature and $V_d$ is the droplet volume in mL (0.05ml) (Figure 2).¶
¶

Next, Poisson statistics were used to calculate cumulative INP concentrations from the frozen fraction data.

**Moved down [2]:** Note that we chose to report our correlation analysis using $T_{50}$ as the temperature where *FF* = 0.5.

concentrations. Thus, the first two wells to freeze out of the 96 wells were omitted for calculating cumulative INP concentrations.

Furthermore, triplicates showed good reproducibility: $T_{50}$ values of sample triplicates fell within 1°C (Figure 3) and standard deviations did not depend on average freezing temperature, consistent with Wright et al. (2013). Refreezing results showed a similar spread in $T_{50}$ values as the triplicates, which suggest that the IN activity of the samples is only minimally affected by freezing (Figure 3). Consistent with the observed spread in triplicate freezing temperatures, the reported temperature uncertainty of DRINCZ is $\pm 0.9$ °C (David et al., 2019). Finally, background corrections for the freezing temperatures were not necessary, as all of the $T_{50}$ values were statistically above the water background of the instrument. Only three of the 88 samples had 75th percentile freezing temperatures overlapping with the mean of the background water: samples 21, 24 and 62 (Table S1). No further data manipulation was done for these samples as the conclusions drawn from these freezing temperatures were the same with or without a correction (Table S1).

[Figure]

**Figure 2: Examples of *FF* curves (blue y-axis), freezing temperature boxplots (bottom of the graphs) and INP concentrations ($n_{sm}$) (red y-axis) generated for two snowmelt samples (from sample #38 and from sample #59, see Table S1). On the boxplot, the thin blue vertical line shows the median and is equal to $T_{50}$, the mean is shown as a blue circle with a concentric dot. The blue box ranges from the 25th to the 75th percentile, whereas the whiskers extend from the 5th to the 95th percentile. Outliers are drawn as blue dots and scattered vertically to avoid overlapping. The figure on the left shows a linear FF curve with overlapping $T_{50}$ and median values on the boxplot, whereas the figure on the right shows a FF curve with a kink with $T_{50}$ and median values differing by at least 1 °C.**

[Figure]

**Figure 3:** Refreezing and triplicate data at different locations show that the variability is within the instrument error of 0.9 °C. The standard deviation of the refreezings (± 0.47 °C) is comparable to the standard deviation of the triplicates (± 0.28 °C). On the boxplots, the blue vertical line shows the median and is equal to $T_{50}$. The mean is shown as a blue circle with a concentric dot and the box ranges from the 25th to the 75th percentile. The whiskers extend from the 5th to the 95th percentile.

**2.4. Physicochemical analyses**

The Swiss alpine snow samples were submitted to chemical analyses in an attempt to correlate parameters to INP concentrations. In particular, total organic carbon (TOC), pH and conductivity measurements were made for all samples, whereas filtering procedures were conducted for a subset of 17 samples.

The total organic carbon was quantified as the non-purgeable organic carbon (NPOC) in solution using a total organic carbon (TOC) analyser (TOC-L CSH, Shimadzu, Japan). This instrument uses a 680 °C combustion catalytic oxidation method, while providing quantification in water samples above 0.1 mg C L$^{-1}$. A detection limit of 4 μg C L$^{-1}$ is achieved through the use of a nondispersive infrared (NDIR) detector. Method sparge time and gas flow were 1.5 mins and 80 mL, respectively. For the NPOC calibration, two calibration solutions were prepared with concentrations of 20 ± 0.2 mg C L$^{-1}$ and 2 ± 0.02 mg C L$^{-1}$ from a potassium phthalate TOC standard solution of 1000 mg C L$^{-1}$ (Sigma-Aldrich).

pH values were measured with a Metrohm pH glass electrode and the values obtained were consistent across all samples, ranging between 5.2 and 6.2. Considering the error on the pH measurements from the unbuffered snowmelt solutions, the pH of all the samples were within error and consequently was not ascribed to any INP variability.

The conductivity of the snowmelt was measured using a handheld conductivity meter (LAQUAtwin COND, Horiba, Japan). It uses small sample volume (less than 0.5 mL) and has an accuracy of ± 1 μS cm$^{-1}$ for conductivity measurements between 0 and 1999 μS cm$^{-1}$. The instrument was calibrated with a 1413 μS cm$^{-1}$ standard solution. Before the measurement was obtained, the sensor was washed three times with nanopure Milli-Q water and then 1 mL of the sample was flushed over the sensor for conditioning.

Finally, to classify the size of INPs in the snow samples, the samples were filtered through cellulose acetate membrane filters with pore sizes of 0.2 and 0.45 μm (514-0063, VWR, USA). In addition, samples were filtered with a 0.02 μm pore filter (Whatman® Anotop® syringe filters, Sigma Aldrich) similarly to Irish et al, (2017). The filtered samples, including molecular biology reagent water blanks, were then measured with DRINCZ. Tests with glass fiber filters with pose sizes of 0.7 μm (SF1300-07,

Moved (insertion) [1]

BGB-Analytik, USA) yielded lower freezing temperatures than with the cellulose acetate membrane filters and were thus not used further for this study.

**3. Results and discussion**

This study investigated the freezing temperatures of 88 snow samples collected over 15 days and 17 different locations during the winter of 2018 in the Swiss Alps. The $T_{50}$ values and freezing temperatures are presented in three sections: physiochemical properties, times series and sampling site characteristics. $T_{50}$ values from the entire dataset ranged between −5.3 °C and −21.6 °C with a mean value of −12.5 ± 4.0 °C (Table S1), all above the DRINCZ instrument's detection limit. We also report correlation analysis using $T_{50}$ as the temperature where $FF = 0.5$ (Table 2). We first use $T_{50}$ values to compare samples to different physicochemical properties of the snowmelt, such as TOC, conductivity and particle size, as well as to collection time within the winter season. We then use freezing temperature boxplots to compare the freezing behavior of samples based on terrain, altitude and snow age. Next, INP concentrations in precipitation were estimated and a parameterization is derived based on a temperature dependence. Finally, we extrapolate the INP concentrations to frozen cloud fractions to infer glaciation temperature and liquid water to ice ratio of mixed-phase clouds over the Swiss Alps in the winter of 2018.

Table 2: Correlation analyses between $T_{50}$ values and different physicochemical and terrain parameters. $R^2$ values characterize the linear relationship between $T_{50}$ and the parameter; the p-value identifies whether the correlation is statistically significant; n represents the number of snowmelt samples used in the correlation.

[revised manuscript text omitted]

15  nucleation over the Swiss Alps, since bacteria are larger than 0.20 µm. However, the role of bacterial fragments or proteinaceous material from sources such as fungi and plant pollen cannot be excluded (Hartmann et al., 2013; Pummer et al., 2012, 2015).

[Figure]

20  **Figure 5** $T_{50}$ **as a function of filter pore size for selected samples. The $T_{50}$ values of the background molecular biology reagent water are shown in black. The grey shaded area above the line represents one standard deviation. The samples were filtered through 0.02, 0.20 and 0.45 µm and the lines connect the different filtrates of the same sample. The $T_{10}$ version of this graph can be found in the supplementary information for further comparison with Irish et al., 2017 and with Wilson et al., 2015.**

**3.2. Time series of $T_{50}$ values**

25  The variability of $T_{50}$ values was observed throughout the measurement period and there was no trend over time (Figure 6). Noticeably, all samples taken in May 2018 had TOC values above the dataset average, and specifically above 1.2 mg C L⁻¹, however the time since their last snowfall was only 1-2 days and so these samples were still relatively fresh (Table S1). Nonetheless, the scatter of $T_{50}$ values indicate no trend overtime of the freezing temperatures of INPs, consistent with a lack of seasonality in

30  INP concentrations measured by a continuous flow diffusion chamber at Jungfraujoch (Lacher et al., 2018a). The absence of time dependency on the INP occurrence within the campaign timeframe indicates that the variability is consistently large throughout the entire timespan investigated.

[Figure]

**Figure 6 Time series distribution of the $T_{50}$ values during the entire field campaign, displayed as months in 2018. TOC values of the samples are shown on a colour scale and conductivity values are shown on a marker size scale.**

**3.3. Spatial, altitudinal, snow age and depth variability of freezing temperatures**

**3.3.1. Spatial heterogeneity of freezing temperatures**

To assess the local variability of INP concentrations spatially, samples from the same site, sampled on the same day were compared based on their distance from each other (Figure 7). The sampling site at Schilt showed consistent freezing temperatures for samples separated by only 5 m, but showed a difference in $T_{50}$ of 4 °C for the sample located 300 m away, within a similar environment. Furthermore, the Sankt Annafirn site also displayed little variability of the median and mean freezing temperatures within a 5 m trajectory on top of the snowpack (Figure 7). Thus, we find that for two sites, snowmelt samples displayed similar INP concentrations within a radius of approximately 5 m. However, 300 m was enough to observe differences in INP concentrations, as was observed at Schilt.

However, for two other sites, Engstligenalp and Weissfluh, the $T_{50}$ values of snowmelt varied by up to 8 °C and 10 °C, respectively. We studied the snowmelt at Engstligenalp due to its location in a flat flood plain of approximately 1 × 2 km in size (Figure S4). Because of the topographical homogeneity of the Engstligenalp site, similar local sources of INPs were expected. However, within the plain of less than 2 km, a difference in $T_{50}$ of 8 °C was observed (Figure 7). Additionally, samples taken on the side of a snow hill at the Weissfluh site, showed a wider spread in freezing temperatures within a distance of only 8 m. Both the Engstligenalp and Weissfluh sites displayed visual evidence of snowdrift, which could explain the large variability observed compared to the Schilt and Sankt Annafirn sites (Figure 7). In fact, wind drift can lead to an extremely heterogeneous snowpack (Gauer, 2001), which could lead to locally heterogenous INP concentrations. Specifically, at Weissfluh, the origin sample and the samples at distances of 2 and somewhat at 4 m had significantly warmer $T_{50}$ temperatures and a narrower spread in freezing temperatures than the snowmelt samples at other distances. The narrow spread suggests that the INPs responsible for the observed freezing in these samples (at origin and at 2 m) were at abundant at these locations, but inhomogeneous across the plain (Figure 7).

[Figure]

**Figure 7: Spatial distribution of freezing temperatures of snow samples collected at four different flat sites, at Schilt on February 24, 2018, at Engstligenalp on April 14, 2018, at Weissfluh on April 15, 2018, and at Sankt Annafirn on April 22, 2018. Weissfluh and Sankt Annafirn are particularly striking since the snow samples collected at these sites show heterogeneity within 1 m distances. All the samples at the same location were collected on the same day. On the boxplots, the blue vertical line shows the median and is equal to $T_{50}$, the mean is shown as a blue circle with a concentric dot and the box ranges from the 25th to the 75th percentile, and the whiskers extend from the 5th to the 95th percentile.**

**3.3.2. Altitude dependency of freezing temperatures**

To investigate the influence of boundary layer and local sources of INPs on snowfall across the Swiss Alps, sampling sites were chosen to cover a broad range in altitude. The sampling sites ranged in altitude between 440 and 3981 m a.s.l. with a median altitude of 2294 m (Table 1), thereby covering a range of sites affected by rare snow events (in Zurich at 440 m a.s.l.) and by eternal snowpack (at Fletschhorn at 3981 m a.s.l.). 
[revised manuscript text omitted]

**4. Atmospheric implications for mixed phase cloud glaciation**

An accurate onset temperature of ice nucleation in supercooled clouds and the subsequent temperature transition to mixed-phase and eventually to glaciated clouds are important factors for determining cloud lifetime and thus radiative forcing. Yet, these onset and transition temperatures are variable and lead to large uncertainties in weather and climate models (Boucher and Quaas, 2013; Prenni et al., 2007; Vergara-Temprado et al., 2018). The glaciation temperature of a cloud further depends on the supersaturation within the cloud and is highly sensitive to updraft velocity (Korolev et al., 2017; Korolev and Isaac, 2003; Korolev, 2008). As ice crystal concentrations in clouds can be several orders of magnitude higher than INP concentrations (Wex et al., 2010), secondary ice processes can play an important role in the evolution of supercooled clouds (Beck et al., 2018; Hallett and Mossop, 1974; Lauber et al., 2018; Mignani et al., 2019; Petters and Wright, 2015). However, the process of ice multiplication is not fully understood, with multiplication factors ranging from one to multiple orders of magnitude (Mignani et al., 2019; Wang, 2013). Nevertheless, it has been proposed that complete glaciation of supercooled clouds can be initiated through ice multiplication with less than 10 INPs per $m^{-3}$ ($c_{air}$) (Crawford et al., 2012; Mason, 1996) by the process of riming and ice splintering (Hallett and Mossop, 1974; Mossop, 1978). Therefore, by choosing a threshold for the number of INPs required to glaciate a cloud, say 10 $m^{-3}$, we can estimate the frozen cloud fractions based on our cumulative INP concentrations (Figure 12).

Specifically, we identified the freezing temperature from all 88 of our samples which corresponded to a calculated $c_{air}$ value of 10 m$^{-3}$ (also equivalent to 25 ml$^{-1}$ of snowmelt ($n_{sm}$). We then sorted these 88 temperatures from warmest to coldest and plotted them in a linear space from 0 to 1, corresponding to a frozen cloud fraction (Figure 12). We repeated this analysis with assumptions of $c_{air}$ values of 20 m$^{-3}$ and of 40 m$^{-3}$ required for cloud glaciation. However, the highest INP concentration from this study did not exceed ~ 200 m$^{-3}$, because of the -22.5 °C limit of detection of DRINCZ. This temperature thus limits the extrapolation of frozen cloud fractions to lower temperatures. Yet, since other studies suggest that hundreds of INPs per m$^{-3}$ can be necessary for cloud glaciation to occur (McCoy et al., 2016), in contrast to the lower limit of 10 m$^{-3}$ by Crawford et al. (2012), we use our parameterization to estimate the frozen cloud fraction from an order of magnitude higher INP concentrations. The change in temperature required for $c_{air}^*$ to increase by an order of magnitude can be calculated as $\Delta T = \frac{ln(10c_{air}^*)}{0.7} - \frac{ln(c_{air}^*)}{0.7}$ and is equal to –3.3 °C. Thus, to extend the frozen cloud fraction using 400 m$^{-3}$ as the INP concentration necessary for cloud glaciation, the frozen cloud fraction at 40 m$^{-3}$ was translated to colder temperatures by –3.3 °C (Figure 12).

Our 50 % frozen cloud fractions fall within the range reported from airborne, ground based, and satellite measurements as summarized in McCoy et al (2016) (Figure 12). Furthermore, these results are remarkably consistent with the typically observed transition zones between supercooled liquid and ice clouds in models and observations (Costa et al., 2017; Henneberg et al., 2017; Lohmann et al., 2016; McCoy et al., 2016; Pithan et al., 2014). Indeed, global circulation models partition liquid and ice in a given atmospheric volume as a monotonic function of temperature, but precipitation and freezing and melting cycles affect the liquid water to ice ratio, thereby modifying the cloud phase transition in often poorly constrained ways (Cesana et al., 2015; McCoy et al., 2015). Thus, temperature remains the dominant effect on influencing liquid cloud fraction (Tan et al., 2014).

[Figure]

**Figure 12.** Frozen cloud fraction as a function of temperature derived from all 88 samples from this study at four different critical INP concentrations necessary for cloud glaciation, namely 10 m$^{-3}$, 20 m$^{-3}$, 40 m$^{-3}$. The frozen cloud fraction was further estimated using the $c_{air}^*$ parameterization to 400 m$^{-3}$ (see text). The aircraft (blue), inferred CALIPSO (red) and ground based lidar (green) temperature ranges for 50% frozen cloud fractions are adapted from (McCoy et al., 2016a).

**Code/Data availability**

Freezing temperature data for all 88 snowmelt samples are available in the supplementary information. Raw data and MATLAB data analysis code are available upon request.

**Moved (insertion) [3]**

[revised manuscript text omitted]